# Origin of giant electric-field-induced strain in faulted alkali niobate films

Moaz Waqar [1,2,3,9], Haijun Wu [4,9], Khuong Phuong Ong [5], Huajun Liu [2], Changjian Li[1], Ping Yang[1,6], Wenjie Zang[1], Weng Heng Liew[2], Caozheng Diao [6], Shibo Xi [7], David J. Singh [8], Qian He [1], Kui Yao [2,3✉], Stephen J. Pennycook [1,3✉] & John Wang [1,2,3✉]

A large electromechanical response in ferroelectrics is highly desirable for developing high-performance sensors and actuators. Enhanced electromechanical coupling in ferroelectrics is usually obtained at morphotropic phase boundaries requiring stoichiometric control of complex compositions. Recently it was shown that giant piezoelectricity can be obtained in films with nanopillar structures. Here, we elucidate its origin in terms of atomic structure and demonstrate a different system with a greatly enhanced response. This is in non-stoichiometric potassium sodium niobate epitaxial thin films with a high density of self-assembled planar faults. A giant piezoelectric coefficient of ∼1900 picometer per volt is demonstrated at 1 kHz, which is almost double the highest ever reported effective piezo-electric response in any existing thin films. The large oxygen octahedral distortions and the coupling between the structural distortion and polarization orientation mediated by charge redistribution at the planar faults enable the giant electric-field-induced strain. Our findings demonstrate an important mechanism for realizing the unprecedentedly giant electro-mechanical coupling and can be extended to many other material functions by engineering lattice faults in non-stoichiometric compositions.

[1] Department of Materials Science and Engineering, National University of Singapore, Singapore 117574, Singapore. [2] Institute of Materials Research and Engineering (IMRE), A*STAR (Agency for Science, Technology and Research), Singapore 138634, Singapore. [3] Integrative Sciences and Engineering Programme, National University of Singapore, Singapore 119077, Singapore. [4] State Key Laboratory for Mechanical Behavior of Materials, Xi'an Jiaotong University, Xi'an, China. [5] Institute of High Performance Computing, A*STAR (Agency for Science, Technology and Research), Singapore 138632, Singapore. [6] Singapore Synchrotron Light Source (SSLS), National University of Singapore, Singapore 117603, Singapore. [7] Institute of Chemical and Engineering Sciences, A*STAR (Agency for Science, Technology and Research), Singapore 627833, Singapore. [8] Department of Physics and Astronomy and Department of Chemistry, University of Missouri, Columbia, MO 65211, USA. [9] These authors contributed equally: Moaz Waqar, Haijun Wu. ✉email: k-yao@imre.a-star.edu.sg; stephen.pennycook@cantab.net; msewangj@nus.edu.sg

There is an extensive ongoing quest for piezoelectrics with a large electromechanical response. This is especially the case for lead-free oxide systems with environmental and regulatory drivers[1]. A common approach is to control oxide perovskite ferroelectric compositions in relation to morphotropic phase boundaries (MPBs), where a large electromechanical coupling effect can be realized[2–4]. This approach involves the engineering of chemical composition to have competing polar phases with similar energies but different structures, leading to a large strain with polarization rotation under external electric field[1–3]. Nanoscale structural and chemical heterogeneity at MPBs plays major roles in achieving a large piezoelectric response by flattening the thermodynamic energy surface[4,5]. Delicate control of complex stoichiometry, with the aid of external dopants, is often crucial for achieving the desired multiphase coexistence and local heterogeneity. Precise control of stoichiometry is more difficult in ferroelectric thin films compared to bulk due to the increased loss of volatile and diffusive species, especially when the film preparation involves high-temperature growth or annealing[6]. Furthermore, substrate clamping effects can result in a shift[7] or even the absence[8] of an MPB in a ferroelectric thin film. Therefore, alternate strategies are needed. In this regard, a large improvement in piezoelectric response has been observed in perovskite oxide ferroelectric crystals and ceramics by chemically and mechanically induced nanoscale defects such as vacancies, dislocations, and precipitates[9–11]. In ferroelectric oxide thin films, it has been shown that the defects associated with deviations from ideal stoichiometry can cause structural and polarization inhomogeneity resulting in a variety of strain and polar states[12,13]. These can be controlled by film growth conditions without requiring complex compositions or extensive chemical doping[13,14]. Recently, a giant electromechanical coupling was observed in a sodium niobate thin film that contained extended lattice defects related to stoichiometry[15,16]. These lattice defects result in structural and polar heterogeneity responsible for the large response as shown by mesoscale simulations. However, a fundamental understanding of the physical mechanism in relation to the atomic structure of the extended defects is essentially required so that these defects can be controlled and films with improved electromechanical coupling can be designed.

Here, we report a giant electromechanical response in potassium sodium niobate epitaxial thin films with a high density of planar faults (PFs) induced by non-stoichiometry. The detailed structural analysis provides the connections among composition, structure, and performance at the atomic scale supported by first-principles calculations. An effective piezoelectric coefficient $d^*_{33,f}$ of ~1900 pm V$^{-1}$ at 1 kHz frequency is obtained in this work, which is almost double the recently discovered giant piezoelectric response of 1098 pm V$^{-1}$ in NaNbO$_3$ composition[15]. A huge electric-field-induced reversible strain of 5.6% is achieved at 100 Hz with an electric field of 83.3 kV cm$^{-1}$. We use atomically resolved microscopy and various spectroscopic techniques to elucidate the structure of the PFs. This, in relation to the first-principles calculations, establishes their relationship with the multiple local polarization states at the nanometer scale. Our analyses of the electromechanical, dielectric, and conductive responses as functions of voltage, temperature, and frequency clarify the role of PFs in causing the giant electric-field-induced strain. We show that the ionic displacements at the PFs in response to the charge-migration-mediated polarization rotation under an external electric field can well account for the giant $d^*_{33,f}$ and electric-field-induced strain in the faulted alkali niobate films.

## Results and discussion

**Structure and electromechanical properties**. We deposited non-stoichiometric $(K_xNa_{1-x})_yNbO_{3-z}$ epitaxial thin films by a sputtering process on (001) oriented Nb-doped SrTiO$_3$ (Nb:STO) single-crystal substrate with increasing alkali deficiency (see Methods). The formation of Nb anti-site defects in alkali niobates becomes favorable in an alkali-deficient environment[15,17] and their self-assembly can result in the formation of nano-pillar regions (NPRs). These are shown in Fig. 1a, which is a plan-view high-angle annular dark-field (HAADF) scanning transmission electron microscopy (STEM) image of the thin film. The density of PFs (as stated in Fig. 1a–c) can be controlled by tuning the (K + Na)/Nb ratio in the films (see Methods), which increases with decreasing (K + Na)/Nb ratio as shown in Fig. 1b, c. At the same time, perovskite octahedral rotations that compete with ferroelectricity are inhibited by the Na/K ionic size mismatch. The electric-field-induced strain in these films was measured at different voltages and frequencies using a scanning laser vibrometer (LSV) and the $d^*_{33,f}$ was determined from the measured strain[18] (Supplementary Fig. 1, Fig. 1d, f). The $d^*_{33,f}$ increases from 914 pm V$^{-1}$ to 1909 pm V$^{-1}$ for thin films with the PF densities of 0.13 nm$^{-1}$ and 0.18 nm$^{-1}$, respectively, as measured at an electric field of 83.3 kV cm$^{-1}$ and a frequency of 1 kHz (Fig. 1d). The maximum $d^*_{33,f}$ values reported here are limited by the breakdown voltage of the films and the absence of saturation in $d^*_{33,f}$ values indicate the possibility of even larger $d^*_{33,f}$ values at higher electric fields. Importantly, the films show a reversible electric-field-induced strain with minimal hysteresis (Supplementary Fig. 2).

A $d^*_{33,f}$ of 1909 pm V$^{-1}$ was obtained from $(K_xNa_{1-x})_yNbO_{3-z}$ thin film with a PF density of 0.18 nm$^{-1}$ (hereafter PF-KNN) at 1 kHz. This is the highest value ever obtained among piezoelectric thin films and is approximately 1.7 times the previously obtained giant $d^*_{33,f}$ in Na-deficient NaNbO$_3$ thin films[15], and four times that of the best Pb(Zr,Ti)O$_3$ (PZT) piezoelectric thin films[19] (Fig. 1e). In contrast, we obtained a $d^*_{33,f}$ of only 42.5 pm V$^{-1}$ in the stochiometric $K_{0.37}Na_{0.63}NbO_3$ (hereafter KNN) thin film measured at 83.3 kV cm$^{-1}$ and 1 kHz (Supplementary Fig. 3). We obtained a giant electric-field-induced reversible strain of 5.6% at 100 Hz in the film with the highest PF density measured at 83.3 kV cm$^{-1}$, which corresponds to a $d^*_{33,f}$ of 6722 pm V$^{-1}$ (Fig. 1f). Such a large macroscopically measured voltage-induced strain surpasses even the large values obtained in relaxor ferroelectrics (Fig. 1g). Compared to Na-deficient NaNbO$_3$ thin films[15], the higher electromechanical response in PF-KNN films can be attributed to the introduction of K, as well as the relatively higher density of PFs in PF-KNN film (see Supplementary Section I for further discussion).

**Atomic-scale characterization of planar faults**. Evidently, the controlled growth of alkali-deficiency-induced extended defects can be utilized as an effective strategy to enhance the electromechanical activity of alkali niobate films. However, the key structural features of these defects driving the observed giant electromechanical response in these films as well as the underlying physical origin at the atomic scale are not yet understood. Using PF-KNN film with the PF density of 0.18 nm$^{-1}$ as the platform, we perform an in-depth analysis to pinpoint those key structural, chemical, and physical mechanisms responsible for the observed electromechanical enhancement. Our X-ray diffraction (XRD) analysis confirmed that the PF-KNN film exhibits a crystalline tetragonal structure (P4mm symmetry) with a $c/a$ ratio of 1.037 (Supplementary Fig. 4). We used STEM to resolve the structural features of the film at the atomic level. The results show that the PF-KNN film has a thickness of approximately 300 nm and that the PFs are homogeneously distributed throughout the observable region of the film (Supplementary Fig. 4). These PFs originate from the film-substrate interface and take the form of

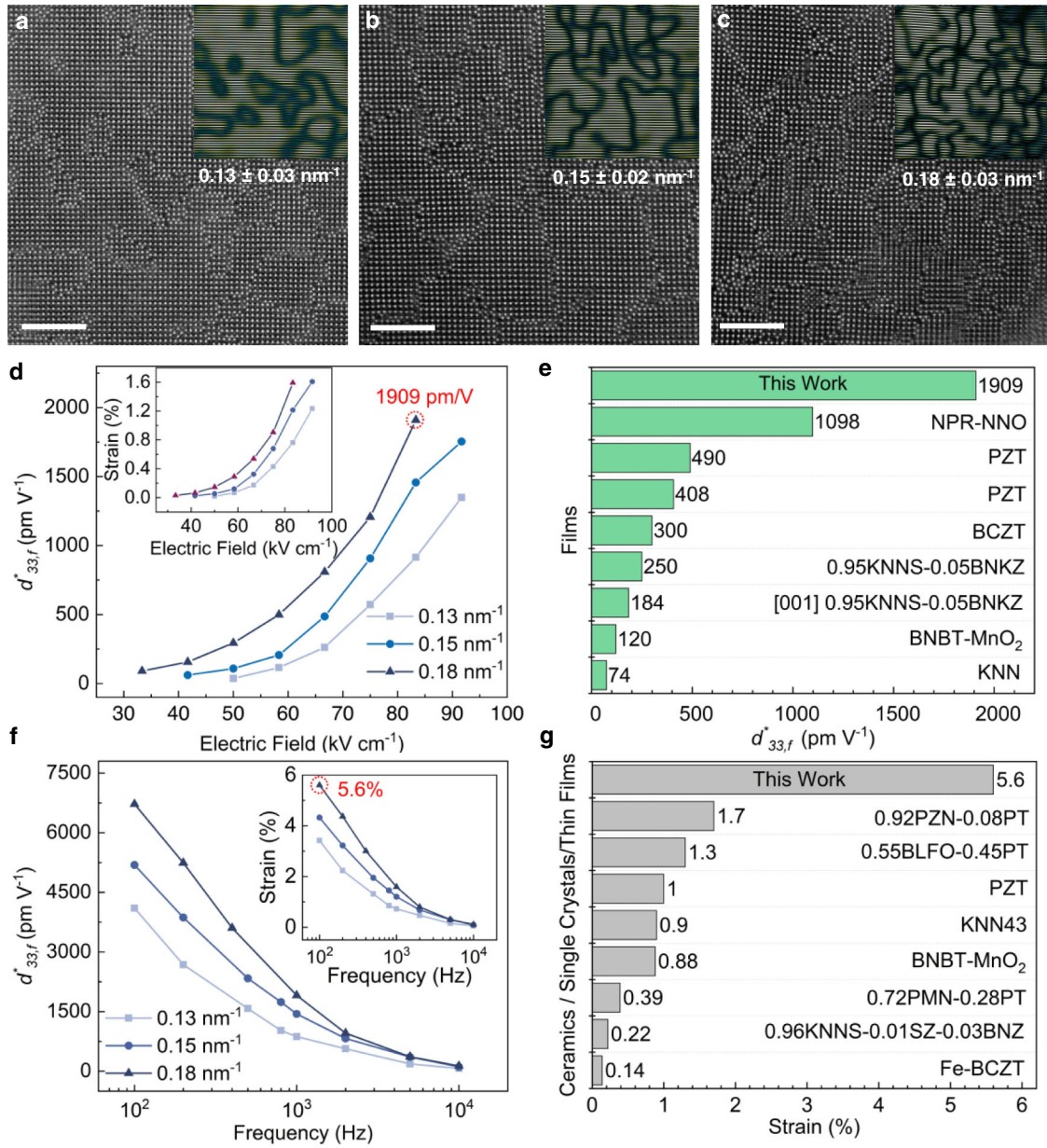

**Fig. 1 Effect of planar fault density on the electromechanical response of the PF-KNN thin film.** Plan-view HAADF-STEM images of thin films with the PF densities of **a**, $0.13 \pm 0.03$ nm$^{-1}$ **b**, $0.15 \pm 0.02$ nm$^{-1}$ and **c**, $0.18 \pm 0.03$ nm$^{-1}$ (scale bars equal 5 nm). **d** Variation of $d^*_{33,f}$ with the applied electric field in films having different PF densities measured at 1 kHz frequency (Inset shows the corresponding electric-field-induced strain). **e** $d^*_{33,f}$ obtained in this work at 1 kHz frequency in the PF-KNN film compared with those of previously reported lead-based and lead-free piezoelectric thin films (Supplementary Table 1 for detailed comparison). **f** Variation of $d^*_{33,f}$ with the excitation frequency in films having different PF densities at the applied field of 83.3 kV cm$^{-1}$ (Inset shows the corresponding electric-field-induced strain). **g** Electric-field-induced strain obtained in this work compared with those of previously reported lead-based and lead-free piezoelectric single crystals, bulk ceramics, and thin films (Supplementary Table 2 for detailed comparison).

extra Nb layers displaced in ⟨100⟩, ⟨010⟩, or ⟨110⟩ directions (Fig. 2a, highlighted by arrows). To further confirm the nature of the PFs, we prepared plan-view samples for STEM imaging (Fig. 2b) and determined these PFs as $a/2[\bar{1}\bar{1}0](100)$ or $a/2[1\bar{1}0](010)$ type stacking faults ($a$ is the lattice constant of a cubic cell). However, a slight vertical mismatch (in [001] direction), $\alpha$, between the faulted regions is observed (see Fig. 2a). Density functional theory (DFT) calculations with the observed structure show that the PF with a slight vertical distortion $\alpha$ is more stable than the perfect $a/2[\bar{1}\bar{1}0](100)$ type stacking fault (see Methods, Supplementary Fig. 5). The DFT results also show that the distance between NbO$_2$-NbO$_2$ planes at the PF interface increases from 2 Å to approximately 2.6 Å

(Supplementary Fig. 6). This is in very good accord with the experimental observations.

As a consequence of these structural distortions, the oxygen octahedra, which are edge-shared between the NbO$_2$ planes at the PF interface, are considerably deformed in both the in-plane and out-of-plane directions, with largely modified Nb-O-Nb bond angles and bond lengths (Fig. 2c, Supplementary Fig. 6). The larger size of the K ions, compared to Na, constrains the O positions, disfavoring octahedral rotation and favoring polar distortions[20]. These oxygen octahedral distortions (OODs) further cause the rumpling of oxygen atomic planes (denoted here as $\beta$) adjacent to the PFs as can be seen in Fig. 2d. This rumpling has a magnitude of up to 0.47 Å, which extends further

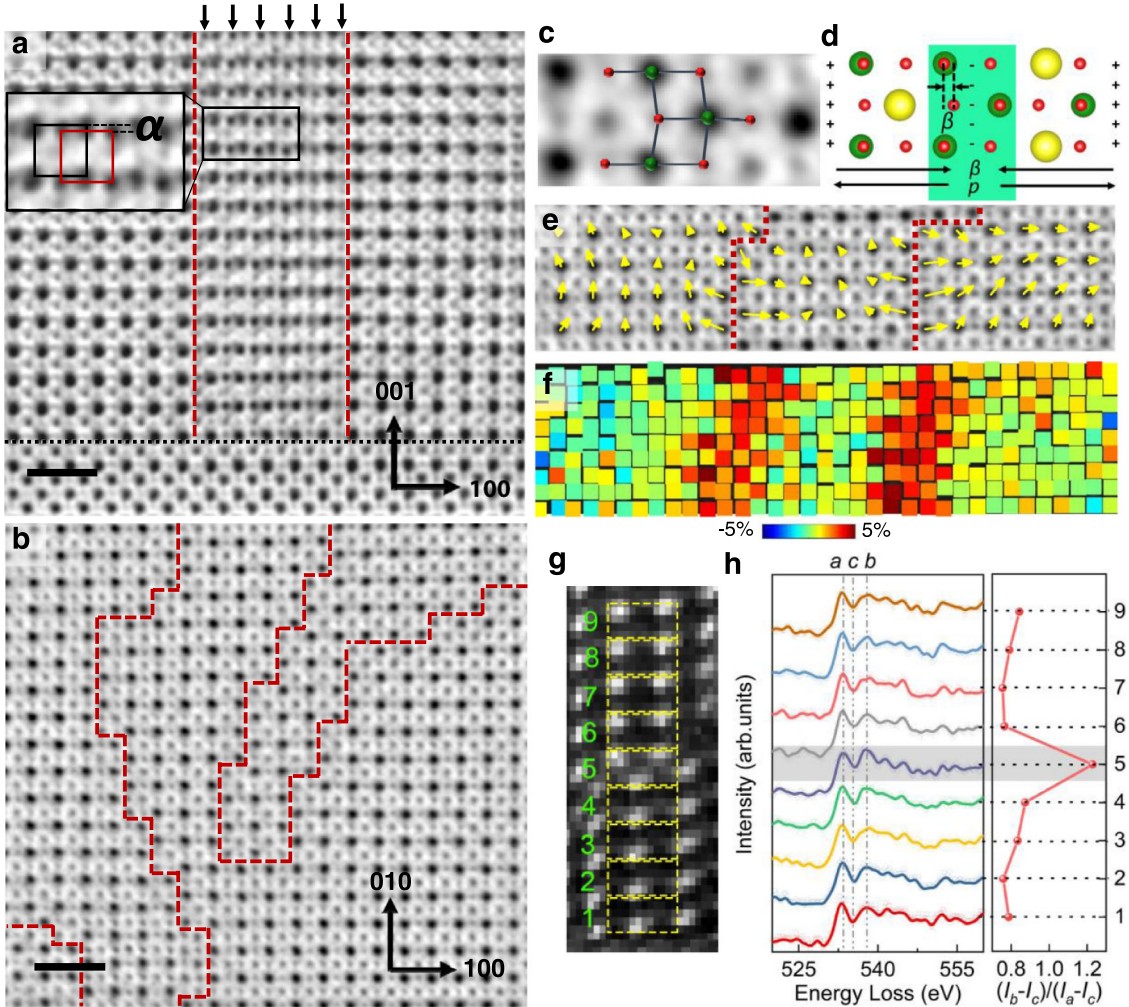

**Fig. 2 Structural and spectroscopic analyses of the PF-KNN thin film. a** An ABF image of thin-film cross-section at the thin film-substrate interface. The arrows point towards the projection of the faults appearing as extra Nb atomic columns. The red dashed and black dotted lines highlight the region containing the planar defects and the film-substrate interface, respectively. The inset shows the vertical mismatch '$\alpha$' between the normal and faulted Nb columns. The scale bar equals 1 nm. **b** A plan-view ABF image of the PF-KNN thin film. The red dashed lines highlight the planar faults. The scale bar equals 1 nm. **c** A magnified ABF image highlighting the OODs and **d**, DFT simulated structure of a PF observed from the plan-view of the film. Green, red, and yellow atoms represent Nb, O, and Na atoms, respectively. Oxygen rumpling '$\beta$' towards the PF center induces a polar displacement or a dipole moment, $p$, pointing away from the PF. **e** ABF image of thin-film plan-view showing two planar faults (highlighted by dotted lines) parallel to each other overlaid with its corresponding in-plane polarization vector ($\delta_{Nb\_xy}$) map. **f** The strain ($\varepsilon_{xx}$) variation across the region shown in **e**. **g** Annular dark-field (ADF) image of the PF region and **h**, the energy loss curves obtained from different regions marked in **g**.

into the nominal perovskite matrix (Supplementary Fig. 7). Such distortions at the PF can be attributed to the decrease in the ionic charge of Nb atoms due to charge redistribution and the modified Coulomb interactions at the PFs caused by the disruption in local symmetry[21] as further detailed in the next section.

**Evidence of localized charged at the PFs**. We confirmed the decrease in Nb valence from +5 by the X-ray photoelectron spectroscopy (XPS) results obtained for the PF-KNN film, which is well supported by our XAS measurements and first-principles calculations (Supplementary Figs. 8 and 9). To further ascertain that such lowering of Nb valence occurs at the PF interface, we used atomically resolved energy-loss near-edge spectroscopy (ELNES) at the O K-edge (Fig. 2g, h). We calculated the factor $\gamma = (I_b - I_c)/(I_a - I_c)$ ($I$ = intensity where $a$, $b$ and $c$ are the peaks labeled in Fig. 2h). This factor qualitatively measures the Nb valence in the vicinity of PFs which should specifically increase with a decrease in Nb valence[22], similar to the O K-edge obtained

from the XAS measurements (See Supplementary section II). It is clear from Fig. 2h that $\gamma$ increases at the PF, confirming the lowering of Nb valence at the PF interface. Thus, the observed structural distortions at the PF interface are coupled with variations in the chemical states of Nb ions. This variation is likely facilitated either by band bending or electron transfer from neighboring oxygen atoms. This is in agreement with the ELNES and XAS results which suggest that the nominally empty $4d$ orbitals of Nb atoms become partially occupied at the PF interface (See Supplementary Fig. 8 and Supplementary Section II). Furthermore, the AC impedance analysis of PF-KNN film shows a low frequency (~100 Hz) relaxation with activation energy, $E_a$, of 0.33 eV indicating the existence of electrons as localized charges possibly in the polaronic form and not as itinerant charge carriers (Supplementary Fig. 10). This is also reflected in the AC conductivity analysis where the $E_a$ of conduction in PF-KNN film is around 0.41 eV in comparison to the 0.02 eV obtained for stoichiometric KNN film (Supplementary Fig. 11), which also confirms the presence of localized charged carriers in PF-KNN

film (See Supplementary Section III for detailed discussion). In ferroelectric materials, such localization is caused by the self-trapping of electrons in potential wells created by local elastic distortions[23]. Hence, in our case, the electronic charge is localized around the PFs due to the presence of an anisotropic elastic strain field caused by the OODs[24,25]. Our Generalized Gradient Approximations for Coulomb interaction potential (GGA + U) show that the electron correlation has no significant effects on the electronic properties suggesting the electron localization to have purely lattice character (see Methods, Supplementary Fig. 9). Accordingly, we conclude that the delicate balance between the elastic distortions and localized charge at the PFs govern the observed in-plane OODs in the PF-KNN film.

**Polarization mapping and domain structure**. In order to understand the coupling of OODs with the surrounding lattice polarization, we analyzed the plan-view annular bright-field (ABF) image of three regions separated by two parallel PFs (Fig. 2e). The displacement of perovskite *B*-site atoms with respect to the corner *A*-site atoms provides an indication of the polarization in displacive ferroelectrics that is directly accessible in STEM measurements and can be compared with DFT calculations[26]. We calculated the in-plane shift of Nb ions with respect to K/Na atoms ($\delta_{Nb\_xy}$) and refer to it as the in-plane polarization vector (see Supplementary Section IV for polarization mapping at the PF interface). Interestingly, the polarization vectors of unit cells adjacent to the PFs point away from the PFs displaying a tail-to-tail configuration similar to charged domain walls which support the presence of charge on the PFs[27] (Fig. 2e). As an additional verification of polarization orientation, we also calculated the in-plane shift of Nb ions with respect to the center of oxygen octahedra ($\delta_{NbO\_xy}$), which although less accessible experimentally, is an important confirmation of the polarization direction. The result confirms the formation of outward dipole moments pointing away from the PFs proportional to the magnitude of $\beta$ (Supplementary Figs. 7 and 12). The PFs also cause a local in-plane strain due to structural mismatch at the PFs (Fig. 2f, Supplementary Fig. 13).

To further visualize the effect of PFs on the domain structure and local symmetry in PF-KNN film, we performed large-area polarization mapping (See Supplementary Section IV). Based on the results, the local strain and built-in electric field present at the dense population of PFs appear to hinder the formation of long-range ferroelectric ordering resulting in nano-regions of complex polarization states enclosed within the PFs. Furthermore, the local chemical and structural heterogeneity at the PFs result in low symmetry phases (See Supplementary Section IV and Methods) in the vicinity which provide a continuous polarization rotation path from the in-plane direction near the PFs to the out-of-plane direction away from the PFs (Supplementary Figs. 14 and 15). These observations show that the interplay between the OODs and the localized charge at the PFs results in the formation of in-plane dipoles which strongly affect the polarization states in the adjacent region.

**Mechanisms for giant electromechanical strain**. Based on the results, the structural and polarization inhomogeneity caused by the PFs in the PF-KNN thin film can be schematically illustrated, as shown in Fig. 3a. The tail-to-tail polarization at the PFs suggests that the local charge redistribution plays a major role in the observed polarization orientations in the vicinity of the PFs. Figure 3b, c depict the structural distortions adjacent to the PFs showing large variations in Nb-O bond lengths along with the in-plane dipole, as calculated by DFT. Large off-centering of Nb ions relative to the oxygen octahedra can be observed immediately

adjacent to the PF due to the reduced ionic charge of Nb[21]. This corresponds to a dramatically enhanced intrinsic electromechanical response including remarkably large lattice polarization adjacent to the PF interface based on the theoretical estimation (see Supplementary Section V). However, our analysis suggests that the intrinsic piezoelectric response cannot account for the observed giant electric-field-induced strain without the involvement of extraordinarily large electrostrictive effects or other extrinsic factors. Polarization rotation between different crystallographic directions and lattice-shearing-induced phase transformations can result in a large electric-field-induced strain in the vicinity of PFs[28–31]. However, under an applied electric field, the reorientation of the in-plane component of polarization in the vicinity of PFs further involves a competition between the applied out-of-plane electric field and the internal in-plane dipole caused by the localized charge and the OODs. A polarization reorientation as shown in Fig. 3d, hence, requires the destabilization of electrical and elastic boundary conditions at the PFs. Based on the experimental evidence provided in the next section, we argue that this polarization reorientation is facilitated and mediated by the electric-field-induced migration of localized polaronic charges along the PFs. The corresponding charge redistribution leads to large ionic displacements in association with the variation of the Nb valence and out-of-plane distortion of oxygen octahedra resulting in a large electric-field-induced strain in the vicinity of PFs. When the field is removed, the original structural configuration reinstates the competing Coulombic repulsion and electron-lattice interactions at the PF interface. Based on this model, we conclude that the large electromechanical response primarily originates from the vicinity of the PFs, and hence, a larger electric-field-induced strain is expected to result from the thin film with a higher density of PFs. This is in accordance with the experimental results shown in Fig. 1a

To validate the correlation between charge migration and large ionic displacements, we investigated the piezoelectric and electrical properties of the films at elevated temperatures, since the localized charge can be activated thermally. From Fig. 1d, we know that the room temperature $d^*_{33,f}$ of PF-KNN film is strongly dependent on the applied electric field. This is in contrast with the conventional piezoelectric response which shows almost no change with the applied field or increases linearly such as in the case of the stoichiometric KNN film (Supplementary Fig. 3). The non-linearity of $d^*_{33,f}$ in the PF-KNN film decreases with the increase of measuring temperature, and relatively higher $d^*_{33,f}$ can be achieved at lower applied fields (Fig. 4a). Such a phenomenon can be well explained by the polaron migration. At low electric fields (Region I in Fig. 4a), there should be almost no or limited activation of polaronic charge at room temperature (RT). This then results in almost no polarization reorientation near the PFs, hence a low $d^*_{33,f}$ of 210 pm V$^{-1}$ is obtained. However, when the temperature is increased to 125 °C, sufficient energy is available for the polaronic charge to overcome their activation barrier $E_a$ even at low voltages and as a result $d^*_{33,f}$ increases to around 697 pm V$^{-1}$ at 125 °C. In contrast, at higher voltages (Region II in Fig. 4a), the energy is sufficient enough for the polaronic charge to overcome $E_a$, the temperature, therefore, has almost no effect on the effective piezoelectric coefficient. This can be captured in Fig. 4b, where the high-temperature electric-field-induced strain in PF-KNN film follows the Arrhenius law, supporting the activation mechanism basis of the electromechanical response. The high-temperature electromechanical behavior of PF-KNN film is in stark contrast with the stoichiometric KNN film which shows a decrease in the $d^*_{33,f}$ at high temperatures and the electric-field-induced strain does not follow the Arrhenius law (Supplementary Fig. 16). This further underpins the presence of

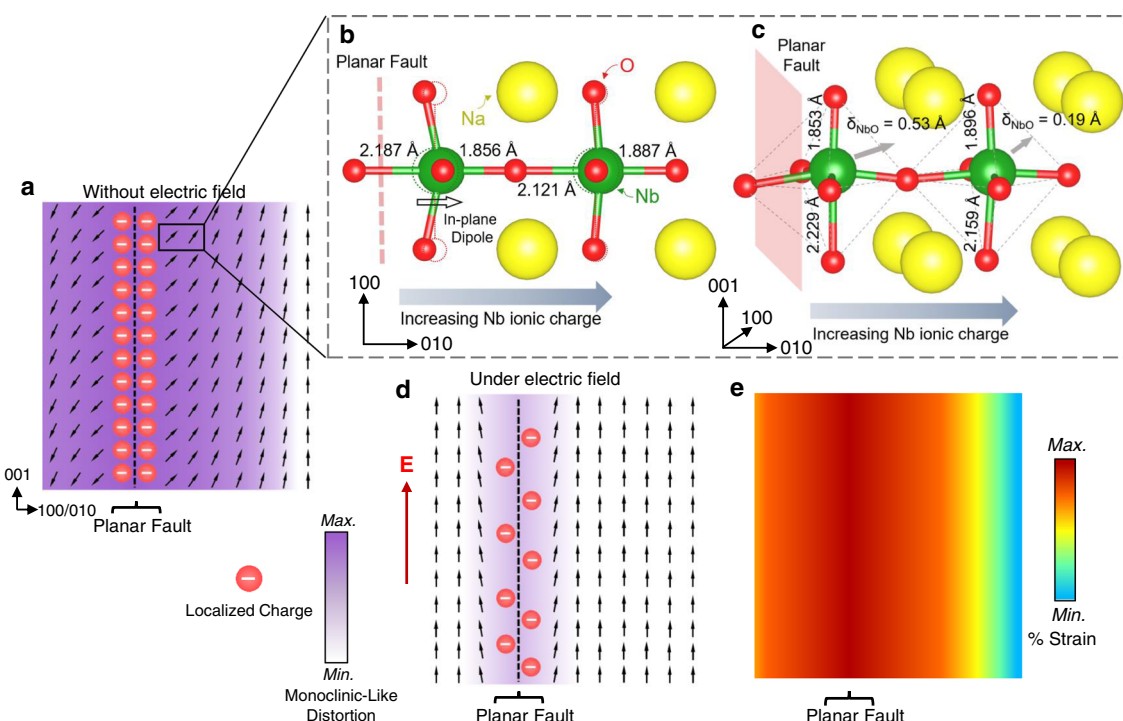

**Fig. 3 Phenomenological model for the large electric-field-induced strain. a** Schematic of PF-KNN thin film cross-section showing polarization states without an external electric field. The localized charge and strain cause symmetry lowering distortion resulting in the in-plane rotation of polarization in the vicinity of PFs and the magnitude of this distortion decreases with increasing distance from the PFs. DFT model showing the **b**, in-plane and **c**, out-of-plane structural distortions adjacent to the PF. **d** Schematic of a PF-KNN thin film cross-section showing polarization states under an electric field, **E**. The polarization aligns with the direction of **E** which is facilitated and mediated by the migration of polaronic charges. Charge migration destabilizes the structure at the PFs including the variation in the Nb valence and lattice deformation due to the electron-charge-induced Coulombic forces. When the field is removed, the polarization configuration returns to its original form as in **a** due to the coupling of polaronic charge with their trapping centers at the PFs. The ionic displacements at the PFs cause large reversible strains in the vicinity of PFs as depicted in **e**.

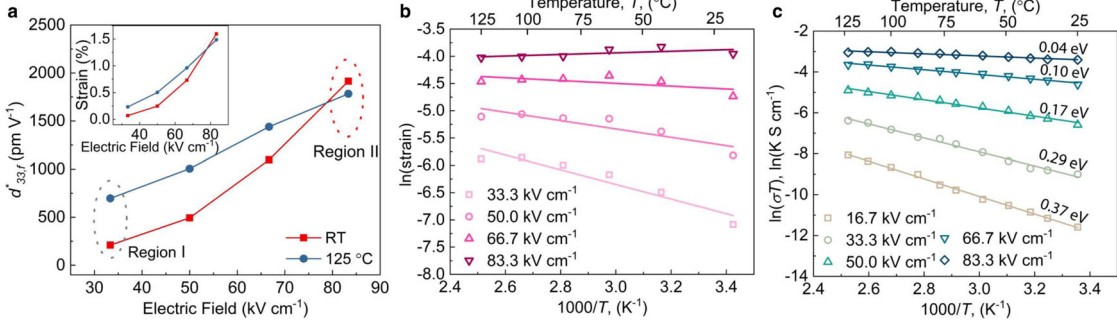

**Fig. 4 Electrical characterization of defects in PF-KNN thin film. a** Variation in $d^*_{33,f}$ (and strain in the inset) measured at room temperature and 125 °C. **b** Variation in the film strain with temperature under different external electric fields presented in Arrhenius formalism. **c** Variation in the dc conductivity of PF-KNN film with the temperature under different external electric fields presented as Arrhenius plot. All the piezoelectric measurements were done at 1 kHz.

fundamentally different physical mechanisms responsible for large electromechanical strain in PF-KNN film compared to that of conventional piezoelectric thin films.

To provide further corroboration to the model, we calculated $E_a$ for polaron migration by measuring the conductivity under different DC electric fields and temperatures as shown in Fig. 4c. $E_a$ obtained at the lowest electric field (corresponding to 16.7 kV cm$^{-1}$) is 0.37 eV which is quite close to the value of 0.33 eV obtained from the impedance relaxation (Supplementary Fig. 10) and 0.41 eV obtained from AC conductivity analysis (Supplementary Fig. 11). These results strongly support the presence of charge localization in the PF-KNN film. On the contrary, the $E_a$ obtained

in the case of stoichiometric KNN is only 0.01 eV which indicates the absence of a thermally activated conduction. With the increasing electric field, the $E_a$ of conduction in PF-KNN film decreases until it becomes close to zero at high electric fields. Hence, a higher concentration of polaronic charges becomes mobile at a higher electric field (see Supplementary Section III). This results in a localized conductivity along the PFs which mediates the electromechanical response of the PF-KNN film. This explains both the temperature-dependent behavior of $d^*_{33,f}$ as well as its strong dependence on the applied voltage.

A strong frequency-dependence of $d^*_{33,f}$ was also observed in PF-KNN film as can be seen in Fig. 1f. In the literature,

frequency-dependent piezoelectric non-linearities are attributed to different underlying mechanisms. For example, the pinning of non-180° domain walls by randomly distributed defects results in a logarithmic increase in the piezoelectric response with the decreasing frequency[32]. In ferroelectric heterostructures or composites, the Maxwell-Wagner relaxation might be observed due to the difference in conductivities of the individual components[33,34]. In some cases, piezoelectric relaxation may also be observed at very low frequencies due to the irreversible movement of conductive domain walls under an external electric field[35]. While the presence of such mechanisms cannot be ruled out completely in PF-KNN film, they cannot account for more than an order of magnitude increase in the $d^*_{33,f}$ at the low frequencies observed in our case. A low-frequency relaxation of the electrostrictive response of the same order has been observed in oxygen-deficient oxides[36,37]. However, we did not observe the presence of such large amounts of oxygen vacancies in our sample (see Supplementary Section VI). The frequency dependence of $d^*_{33,f}$ can, in fact, be well fitted with the non-ideal Debye equation suggesting that the piezoelectric relaxation should be associated with the reorientation of charged defects (polaronic charge) under the alternating field[38,39] (Supplementary Fig. 17). Since the relaxation frequency of small polarons is in the range of $10^0 - 10^3$ Hz[17], the fitted value of piezoelectric relaxation frequency (~160 Hz) is in the expected range. The polaronic charge is bound by the trapping potentials, and unlike conduction electrons, polaron transport is by hopping with activation barriers[40]. In particular, polaron migration induces structural distortions such as the change of Nb-O bond length as well as the valence change of Nb atoms resulting in a high effective mass. Thus, polaron migration becomes inactive at high frequencies, which results in a significantly reduced $d^*_{33,f}$ of 133 pm V$^{-1}$ with frequency increased to 10 kHz. In contrast, the polaron migration can follow the applied alternating field at low frequencies, hence an extremely large $d^*_{33,f}$ of 6722 pm V$^{-1}$ is obtained at 100 Hz. All these results suggest that the strong frequency-dependence of $d^*_{33,f}$ is related to polaron migration. We did not observe a large variation in the piezoelectric response of the normal KNN film with changing frequency (Supplementary Fig. 3). Hence, the polaron migration-mediated ionic displacement associated with the polarization reorientation at the PFs accounts well for the observed temperature and frequency dependence of piezoelectric and conductivity responses in the PF-KNN film.

The electric-field-induced strain in ferroelectric oxide ceramics is usually limited to 1-2% due to the limited elasticity and results in brittle fracture. However, at reduced dimensions, films can accommodate larger atomic displacements than ceramics resulting in a recoverable strain exceeding the elastic limit of the bulk counterparts[41,42]. In the case of PF-KNN film, the observed recoverable strain of at least 5.6% far exceeds the maximum strain observed in lead-based and KNN-based bulk ceramics reported to date (Supplementary Table 2). We attribute this superior elastic behavior to the periodic structural heterogeneity and broken symmetry caused by the PFs. We simulated the structure of PF-KNN film (unstrained) and increased the out-of-plane lattice parameter, $c$, to up to 5% (strained) using DFT (Supplementary Fig. 18). The relatively small energy difference of 132 meV per formula unit between the strained and unstrained structures is consistent with the fact that a strain of 5% is achievable in the KNN structure with the PFs on the application of an electric field. Furthermore, the structure shows considerable lattice shearing near the planar faults along with large out-of-plane deformation of oxygen octahedral (Supplementary Fig. 18). This is most likely caused by the modified elastic properties in the vicinity of PFs due to the broken symmetry and explains the increased electromechanical response at higher defect density (Fig. 1d, f). The

microscopic structure of the film allows the accommodation of large ionic displacements under the applied field, which then results in a giant strain at low frequencies, e.g. 5.6% at 100 Hz under an electric field of 83.3 kV cm$^{-1}$.

In summary, we have discovered an extremely large electric-field-induced strain in non-stoichiometric potassium sodium niobate thin films with self-assembled high-density planar faults. We demonstrated a giant effective piezoelectric coefficient of approximately 1900 picometer per volt at 1 kHz. Atomic-scale measurements and spectroscopic analyses in relation to DFT calculations reveal that the planar faults consist of large oxygen octahedral distortions with localized polaronic charges that reconfigure the local symmetry and polarization in their surrounding regions. A tail-to-tail polarization with largely enhanced magnitude is observed at the planar fault with continuous polarization rotation between the planar faults and the matrix. The voltage, temperature, and frequency-dependent electric-field-induced strain behaviors suggest that the polaronic charge migration along the planar faults facilitates the large ionic displacements associated with the polarization reorientation under an applied electric field. The resulting out-of-plane shearing and elongation of the unit cells in the vicinity of planar faults facilitated by the structural heterogeneity results in the large observed electric-field-induced strain as supported by the theoretical calculations. The edge-sharing of oxygen octahedra over the planar faults and the multivalent nature of Nb are the main features of the faulted films in our work that play combined roles in the observed electromechanical behavior. The giant piezoelectric coefficient and clarified atomic-scale mechanism in the present work demonstrate the effectiveness of a novel strategy for realizing unprecedentedly giant electromechanical coupling. This strategy can be potentially applied to other material functions by engineering lattice defects using non-stoichiometric compositions.

## Methods

**Thin film fabrication and composition analysis.** $(K_xNa_{1-x})_yNbO_{3-z}$ films were deposited on (001)-oriented 0.5% Nb-doped SrTiO$_3$ (Nb:STO) substrates by RF magnetron sputtering. In the chemical formula, $y$ indicates the (K + Na)/Nb ratio, where a decrease in $y$ indicates an increase in the alkali deficiency and $z$ represents the possible oxygen non-stoichiometry. The deposition was carried out at 680 °C for various durations at a pressure of 3.5 mtorr and an Ar/O$_2$ ratio of 3. Due to the high temperature required for the crystallization of the perovskite phase and the high surface area of thin film, K and Na are highly prone to vaporization. The addition of excess K$_2$CO$_3$ and Na$_2$CO$_3$ in the sputtering target can compensate for the volatilization loss. As we found that the K loss is more rigorous than Na, we used the ceramic target with the composition of $(K_{0.55}Na_{0.50})NbO_3$ having 10% excess K. We achieved near stoichiometric film having the composition of $(K_{0.37}Na_{0.63})NbO_3$ (KNN in the main text) with a reduced target to substrate distance ($d_{ts}$) as compared to non-stoichiometric $(K_{0.26}Na_{0.74})_{0.52}NbO_{3-z}$ film (PF-KNN in the main text with PF density of 0.18 ± 0.03 nm$^{-1}$) deposited with an increased $d_{ts}$. The other two films with planar fault (PF) densities of 0.13 ± 0.03 nm$^{-1}$ and 0.15 ± 0.02 nm$^{-1}$ (Fig. 1a, b in the main text) were grown by adjusting $d_{ts}$, having the chemical formulae of $(K_{0.24}Na_{0.76})_{0.85}NbO_{3-z}$ and $(K_{0.26}Na_{0.74})_{0.60}NbO_{3-z}$ respectively. The chemical compositions of all the thin films were obtained by quantifying XPS data using CasaXPS software.

**PF density measurement.** PF densities were calculated using low-magnification HAADF-STEM images. At least three images per sample with the dimensions of 90 nm × 90 nm obtained from different regions were converted into their inverse fast Fourier transformed (FFT) versions before analysis. 3 × 3 equidistant lines (90 nm in length each) were drawn both horizontally and vertically. The average number of PFs intersected by these lines was divided by 90 to calculate the areal PF density with units in nm$^{-1}$. PF density of 0.1 nm$^{-1}$ should correspond to 1 fault per 10 nm.

**X-ray measurements.** Out-of-plane XRD measurements were conducted using a Bruker Discovery D8 diffractometer with the X-rays penetrating a few micrometers inside the sample. Reciprocal space mappings (RSMs) were obtained using high resolution (HR) synchrotron XRD at the XDD (x-ray diffractometry and demonstration) beamline of Singapore Synchrotron Light Source (SSLS). The lattice parameters were found to correspond to a tetragonal structure, using the

reciprocal-space vector (RSV) method[43]. XPS data were acquired by AXIS Ultra XPS. In order to acquire a signal from the non-oxidized surface, the samples were scraped lightly at the testing area with a diamond scriber right before the measurement. Fitting of XPS data was done using CasaXPS software. XAS was performed at the Nb M-edge and O K-edge at room temperature in the SUV beamline at SSLS[44].

**STEM and energy-loss near-edge structure (ELNES) measurements**. Plan-view specimens were prepared by polishing the samples using a mechanical tripod followed by argon ion milling from the substrate side at 8 kV using Leica EM RES102 Ion Mill. Cross-sectional lamellas were prepared using focused ion beam (FIB) milling (FEI Versa 3D microscope). The samples were thinned using successive milling by 30 kV, 8 kV, and 5 kV ion beams where a 2 kV beam was used for final cleaning. STEM imaging and ELNES measurements were done using a JEOL ARM200F atomic resolution electron microscope equipped with a cold field emission gun, an ASCOR 5th order aberration corrector, and a Gatan Quantum ER spectrometer under an acceleration voltage of 200 kV. 10 – 15 images were taken from a single region and averaged for both HAADF and ABF images which were then average-background-subtraction filtered (ABSF)[45] for improved contrast. Atomic displacements were measured using a prewritten script[46] on MATLAB. Atomic models were made using Vesta software[47]. Due to the atomic number (Z) dependence of HAADF and ABF imaging mode, Nb columns (Z = 41) show higher contrast compared to K/Na (Z = 11/19) and O (Z = 8) columns which are less noticeable.

**Electric-field-induced strain measurement**. Voltage- and frequency-dependent displacement of the thin film samples under an alternating current (AC) electric field was measured using a laser scanning vibrometer (LSV, PSV-400, OFV-3001-SF6, PolyTech GmbH). Au electrodes with a diameter of 200 μm were sputtered on the films followed by the baking at 200 °C for 15 min. Before measuring the electric-field-induced strain, the substrate was clamped to a glass slide by using an adhesive. Both the smaller electrode and substrate clamping can reduce the substrate bending effect and mechanical resonances[48]. A tungsten probe tip was contacted at the edge of the electrode for biasing whereas the substrate (Nb-doped SrTiO₃) was grounded (Supplementary Fig. 1). During the measurement, unipolar alternating current (AC) voltage was applied to the top electrode whereafter a focused laser beam then scanned a region of thin-film including electroded and non-electroded areas measuring both the magnitude and phase of the vibration over the area for both the film covered and not covered by the electrode. As a result, a 3-dimensional (3D) profile was obtained which provides information on the vibration magnitude and phase, over the area covered by the electrode and the surrounding substrate area without electrode coverage (Summplementary Fig. 3). The displacement from non-electroded area i.e., the substrate displacement ($\delta_{sub}$) was subtracted from that of the film under the electrode ($\delta_{film}$) to obtain the thin film deformation by eliminating the substrate bending effect if any. Further details are provided in Supplementary Section VII and can also be found in ref 18.

**Impedance and conductivity analysis**. An impedance analyzer (HP4294A, Agilent Technologies Inc.) with a heating stage was used to obtain impedance measurements at different temperatures. The current measurements for obtaining the conductivity were carried out under different fields at different temperatures using a Keithley 2400 source meter.

**First-principles calculations**. To study the PF-KNN structures, We conducted first-principles calculations within density functional theory (DFT) starting from $K_{0.3}Na_{0.7}NbO_3$ structure using the PBESol generalized gradient approximation (GGA-PBESol)[49]. The calculations were done with the full-potential (linearized) augmented plane wave (FP-LAPW) + local orbital method as implemented in the WIEN2K package[50]. The electronic structure was calculated with the augmented plane wave + local orbital (APW + lo) basis set with APW sphere radii ($R_{MT}$) of 2.16 a.u., 2.24 a.u., 1.68 a.u., and 1.52 a.u. for Na, K, Nb, and O, respectively. The partial waves were expanded up to $l_{max} = 10$ in the APW spheres and an interstitial plane wave cutoff $K_{max}$ defined by $R_{min}K_{max} = 7.0$ was used. The charge density was Fourier-expanded up to $R_{min}G_{max} = 18.0$. Here, $R_{min}$ is the smallest sphere radius, of 1.52 a.u. Local orbitals were used to include the Na 2s and 2p, K 3s and 3p, Nb 4s and 4p semi core states in the valence states. The Brillouin zone sampling was done using a 1 × 8 × 7 mesh for the perfect $a/2[\bar{1}\bar{1}0](100)$ type stacking fault as well as the distorted PF structure. To build the PF-KNN structure, we used the P4mm unit cell with experimental lattice constants, $a = 3.925$ Å, $b = 3.925$ Å, and $c = 4.07$ Å. We built a supercell of 11 × 1 × 1 and removed two NaO/KO layers at the interface then shifted one-half of the supercell (1/2a,1/2b,0) in accordance with the structure presented in Fig. 2e with the final composition of 2 K, 7 Na, 11 Nb and 31 O atoms. For distorted planar faults, we shifted one-half of the supercell (1/ $2a + \alpha,1/2b + \alpha,0$). The new optimized structure at the planar fault has lattice constants of $a = 40.68$ Å, $b = 3.905$ Å, and $c = 4.05$ Å with monoclinic Pm symmetry. All structures were fully relaxed until the Hellmann-Feynman forces were less than 1 mRy per a.u. The result shows that the distorted PF structure is more stable than the perfect stacking fault with an energy difference of $\Delta E = E_{DPF} - E_{PSF} = -31.54$ meV per formula ($E_{DPF}$ is the energy of distorted PF, and $E_{PSF}$ is the energy of perfect stacking fault). The density of states (DOS) was calculated for

both KNN and PF-KNN films. In the case of KNN, the structure was built from NaNbO₃ with a 5 × 1 × 1 supercell by replacing one Na atom with a K atom. The DOS show metallic behavior at the planar interface which explains the large conductivity and leakage at high electric fields. However, the carrier localization at the planar faults due to lattice interactions presents a barrier to the conductivity at low electric fields having an activation energy of ~0.4 eV. To eliminate the possibility of the electron correlation effect at the planar fault, we conducted Perdew-Burke-Ernzerhof (PBE + U) calculations. Specifically, we performed GGA + U calculations with the Hubbard parameter set to a large value U = 0.5 Ry (6.8 eV) as shown in Supplementary Fig. 9. The obtained results still show a metallic state, with the density of states similar to the calculation with U = 0. This suggests that the carrier localization is unlikely to be caused by the Coulomb correlations further supporting our argument of carrier localization having a lattice character.

**Reporting summary**. Further information on research design is available in the Nature Research Reporting Summary linked to this article.

## Data availability
The authors declare that the data that support the findings of this study are available within the article and its Supplementary Information files. All other relevant data are available from the corresponding authors upon request.

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

## Acknowledgements

This research is partially supported by A*STAR, under RIE2020 AME Individual Research Grant (IRG) (Grant No.: A20E5c0086). H.W. acknowledges the financial support from the National Key R&D Program of China (2021YFB3201100), National Natural Science Foundation of China (52172128) and Singapore Ministry of Education Tier 1 grant, R-284-000-212-114, for the Lee Kuan Yew Postdoctoral Fellowship. S.J.P. and J.W. acknowledge support from the Singapore Ministry of Education through a Tier 2 grant MOE2017-T2-1-129 for the research conducted at the National University of Singapore. P.Y. acknowledges the support by SSLS via NUS Core Support C-380-003-003-001. The authors would also like to acknowledge the Singapore Synchrotron Light Source (SSLS) for providing the facilities necessary for conducting the research. The SSLS is a National Research Infrastructure under the National Research Foundation Singapore. M.W. acknowledges the experimental assistance by Q.Q. Ke. K.P.O would like to thank IHPC-A*STAR for financial support and the core-research fund. Theoretical work was supported by the A*STAR Computational Resource Centre using its high-performance computing facilities. H. Liu, acknowledges the support of the A*STAR Career Development Fund (project No. C210812020). Work at the University of Missouri was supported by the U.S. Department of Energy, Basic Energy Sciences, Award DE-SC0019114. The authors also acknowledge the technical support from Polytec South-East Asia Pte Ltd. and the assistance from Mr. Darren Lee.

## Author contributions

M.W. and H.W. contributed equally to this work. M.W. and K.Y. conceived the main idea; M.W. grew the samples; M.W. and H.L. carried out the electric testing; M.W., H.L., W.H.L., and K.Y. analyzed electric testing results; M.W. and H.W. performed aberration-corrected STEM; M.W., C.L., W.Z., Q.H., and S.J.P. analyzed the data obtained from STEM; M.W. and P.Y. performed x-ray diffraction and analyzed the data; K.P.O. performed DFT calculations and built theoretical models with input from M.W. and D.J.S. and analyzed the results; S.X. and C.D. performed XAS measurements. K.Y., S.J.P., and J.W. planned and guided the work; M.W., H.W, K.Y., S.J.P., K.P.O., K.Y., D.J.S., and J.W. wrote and modified the manuscript with input from others. All authors discussed the results and revised the manuscript.

## Competing interests

Moaz Waqar, Haijun Wu, Huajun Liu, Kui Yao, Stephen J. Pennycook and John Wang have filed provisional Singapore Patent 10202003767Q. Other authors declare no competing interests.
