## [Peer Review File · Nature Communications]

Origin of Giant Electric-Field-Induced Strain in Faulted Alkali Niobate FilmsREVIEWER COMMENTS

Reviewer #1 (Remarks to the Author):

M. Waqar et al. demonstrate a small-signal piezoelectric coefficient of ~1900 pm/V at 1 kHz, and a large-signal electromechanical strain of 5.6 % in non-stoichiometric potassium sodium niobate epitaxial thin films with planar faults. Some of the authors have introduced a similar enhancement of the piezoelectric properties already in a 2020 science (DOI: 10.1126/science.abb3209) and nature communications (<https://doi.org/10.1038/s41467-021-231>) publications for NaNbO₃. In the current manuscript the authors focus on explaining the mechanism, utilizing (K,Na)NbO₃ thin films as an example.

The authors find that the density of planar faults can be controlled by tuning the (K+Na)/Nb ratio. With increasing planar fault concentration, an increase of the small- and large-signal piezoelectric response is reported. The enhancement of the piezoelectric behavior is prominent at low frequencies and is highly non-linear with the application of the electric field. The piezoelectric and strain performance is considerably enhanced in comparison to conventional thin films made from KNN or PZT. The authors characterize their films using a combination of advanced transmission electron microscopy, macroscopic spectroscopic analysis (impedance measurements and XPS) and DFT calculations. In summary, the authors explain the high electromechanical strain by self-trapped electrons and highly distorted oxygen octahedra in the vicinity of the self-assembled planar faults.

Innovation in strategies to enhance the piezoelectric performance beyond conventional PPT and MPB approaches are important for future thin film and bulk piezoelectric materials in terms of sustainability. In this frame the current manuscript is interesting and timely.

However, the mechanism of the piezoelectric enhancement, the role of intrinsic piezoelectric properties and domain walls is unclear. To make the manuscript understandable and explain the mechanism to a broader audience this needs to be clarified, the core-message of the mechanism needs to be worked out more clearly and the impact of the work for the broader community needs to be explained. In the current form, I cannot recommend the manuscript for publication. I particularly suggest that the authors focus on the following points:

- 1) The mechanism is unclear. According to my understanding the huge strain somehow originates mainly from the pillars formed by the planar faults. Can the authors please be crystal clear: What is the interplay between local elastic strains, polaronic charges, oxygen octahedra, polarization orientation of the surrounding domains and the applied electric field?
- 2) In the summary (page 15, line 334) the authors explain, that “planar faults provide an intermediate pathway for polarization switching”. No experimental evidence / discussion in the manuscript is available to support this statement. Experimental evidence should be provided, or the statement should be removed.
- 3) Does the piezoelectric response change as a function of time? Many PPT and MPB compositions have the advantage of a low fatigue. How are the novel materials the authors are suggesting performing in comparison.
- 4) What is the domain structure of the thin film and how do the interfaces formed by the planar faults impact the domain structure? Can experimental evidence (i.e. PFM) be provided? The authors discuss that tail-to-tail polarization configurations form, but they do not mention anything regarding domain morphology or the density of domain walls in the vicinity of the planar faults.
- 5) How do domain walls contribute to the piezoelectric response? In MPB compositions contributions from domain walls are dominating. Since the authors discuss their work in this context, I suggest that extrinsic / intrinsic contributions to the piezoelectric performance are discussed, to make it easier for the broader community to understand the mechanism the authors are suggesting.

- 6) The nonlinearity in the piezoelectric response can also originate from ferroelectric domain wall contributions (i.e. work by D. Hall: <https://link.springer.com/article/10.1023/A:1017959111402>). The authors utilize the nonlinearity of the piezoelectric properties to corroborate the polaron migration model. Can the nonlinear behavior also be explained by domain wall contributions? In general a more extended discussion on the contributions of domain walls to the piezoelectric performance is required in the manuscript.
- 7) The authors have published and discussed on the effect of planar faults in NaNbO_3 . In the current manuscript they are using $(\text{K},\text{Na})\text{NbO}_3$. Can the authors explain why the effect is enhanced in $(\text{K},\text{Na})\text{NbO}_3$ in comparison to NaNbO_3 ? The authors state that “the higher electromechanical strain can be attributed to the introduction of K...” (line 115, page 5), the mechanism, however, is not explained.
- 8) Why do the planar faults self-assemble into a nanopillar?
- 9) The localized charges seem to play a prominent role in the mechanism for the piezoelectric enhancement. Can the authors provide experimental proof for the existence of the localized charges? Techniques such as conductive Atomic Force Microscopy should be used to proof the existence of such charge carriers.
- 10) Temperature-dependent displacement (Figure 4b) and temperature-dependent conductivity (Figure 4c) on PF-KNN thin films were used to support the polaron model suggested by the authors. Were similar experiments carried out on the stoichiometric KNN thin film (free of planar faults). Which values for the activation barrier were found for stoichiometric KNN?
- 11) Figure S10a: Was a similar experiment carried out for stoichiometric KNN. Can a peak be observed? Relaxation in the frequency range of 10^0 - 10^3 Hz might have many reasons, including i.e. interfaces: <https://doi.org/10.1063/1.2757098>. Can the authors rule out such contributions, i.e. by performing measurements with different electrode materials or by comparing the results of KNN and PF-KNN?
- 12) Which role does the electric field distribution in the material play? I would expect that due to the presence of charge carriers at the planar faults the electric field is locally decreased (since the conductivity is locally enhanced), which would result in a decreased local piezoelectric response at the planar faults. Can the authors discuss the role of an inhomogeneously distributed electric field? I suggest that the authors discuss the work by Rojac et al. (i.e. DOI: 10.1002/adfm.201402963) and Geneko et al. (<https://link.aps.org/doi/10.1103/PhysRevB.97.144101>) in this context.
- 13) The authors find a tetragonal structure of the thin film (line 146, page 7). According to literature, pure KNN should be orthorhombic. Why is the film of the authors tetragonal?
- 14) The authors state that the symmetry lowering in the vicinity of the planar faults is monoclinic like (line 247, page 12). Based on which experiment do they conclude this?
- 15) Figure 4b: Why is the temperature-dependence of the height change used? The unit “ln(nm)” is not straightforward at all. Wouldn't it make more sense to plot the strain (unitless) instead?
- 16) Figure 4c: The authors plot the DC-conductivity as a function of temperature? Where is this value coming from? I do not find frequency-dependent conductivity measurements, which allow to identify the DC conductivity in the supplementary (compare literature by Lunkenheimer et al. 10.1140/epjst/e2010-01212-5).
- 17) The authors conclude that their mechanism “can be potentially applied to other materials” (page 16, line 341). Can the authors provide examples going beyond thin films?

Reviewer #2 (Remarks to the Author):

The authors reported a record-high piezoelectric coefficient of about 1900 picometer per volt at 1 kHz in non-stoichiometric potassium sodium niobate epitaxial thin films ($(K_x Na_{1-x})_y NbO_{3-z}$) with a high density of self-assembled planar faults. They also used first-principles modelling to elucidate the origin of this giant piezoelectric response in terms of atomic structure. The results are very impressive and deserve publication in some form. However, I have some comments for the authors to address:

1) Fig. S5 shows the optimized crystal structure of $(K_{0.3}Na_{0.7})NbO_3$ with one NaO/KO layer at the interface removed and then the two terminations are shifted by one-half of the supercell to have a "plane fault" (i.e. edge-sharing oxygen octahedra). Compared to the ideal stacking (panel a), the distorted structure (panel b) has lower energy. Fig. S6 shows that the Nb-O bond length at the interface increases from 2 Å to 2.6 Å. Presumably, Fig. S5 and S6 are the same DFT results. However, if one examines the two structures carefully, one finds that in Fig. S5 panel b, the NbO_6 octahedra at the interface have rotations and tilts, while in Fig. S6 panel b, the NbO_6 octahedra do NOT have any rotations and tilts and only Nb atoms are shifted along (001) direction, leading to polarization. Could the authors comment on why the optimized crystal structures in Fig. S5 and Fig. S6 are different?

2) A related comment: Line 154-155, the authors wrote "The larger size of the K ions compared to Na constrains the O positions, disfavoring octahedral rotation and favoring polar distortions". However, from Fig. S5, it seems that there are NbO_6 oxygen octahedral rotations and tilts, in particular at the interface (interestingly, such oxygen octahedral rotations and tilts are absent in Fig. S6). Could the authors also comment on that?

3) In the simulation, the K atoms are highly ordered, which also form a plane, similar to "plane fault". But experimentally, K and Na atoms are randomly alloying. Does the result (Fig. S6) depend on how close the "K plane" is to the interface, whether the two "K planes" are symmetric with respect to the interface and whether the K atoms are ordered or randomly distributed?

4) The DFT calculations focus on the crystal structure but do not show the electronic structure. Could the authors provide the detailed density of states of the $(K_{0.3}Na_{0.7})NbO_3$ thin films with "plane faults" as well as the one without "plane faults"?

5) A related comment: Line 189-196, the authors wrote "confirming the lowering of Nb valence at the PF interface", "This variation is likely facilitated either by band bending or electron transfer from neighboring oxygen atoms", "the nominally empty 4d orbitals of Nb atoms become partially occupied at the PF interface", "these electrons exist as localized charges possibly in the polaronic form and not as itinerant charge carriers". Do the DFT calculations support the above claims, in particular from the electronic structure?

6) In the DFT calculations, presumably the $(K_{0.3}Na_{0.7})NbO_3$ thin films with "plane faults" are insulating. If that is the case, can the authors directly calculate d_{33} and compare it to the experiment? In addition, bulk $(K_{0.3}Na_{0.7})NbO_3$ without "plane faults" is also insulating. Can the authors also calculate its d_{33} and compare it to the one with the "plane faults"?

Reviewer #3 (Remarks to the Author):

The authors report their finding of a high piezoelectric coefficient, approximately 1.9 nm/V, and a large strain of 5.6% induced by electric fields in non-stoichiometric potassium sodium niobate [(K_xNa_{1-x})_yNbO_{3-z}] thin films. The authors further provide an interpretation, based on experimental measurements and first-principles calculations of atomic structures (with an emphasis of the role of planar faults), polarization, and electronic properties, about the large strain induced by electric fields. As piezoelectrics with large electromechanical response is of great interests in industry, this work have potentially engineering applications. The conclusions are supported by both the experimental data and DFT calculations. The manuscript is also well-written. Thus, I would recommend an acceptance provide that the authors can address the following points in a revised manuscript:

1. The PF-KNN films [(K_xNa_{1-x})_yNbO_{3-z}] are the focus of this work. Three films, (K_{0.26}Na_{0.74})_{0.52}NbO_{3-z}, (K_{0.24}Na_{0.76})_{0.85}NbO_{3-z}, and (K_{0.26}Na_{0.74})_{0.60}NbO_{3-z} were obtained for experimental analysis while DFT calculations focused on K_{0.3}Na_{0.7}NbO₃. These films all have a similar relative ratio (approximately 1:3) of the K and Na contents. Did the work involve other films with difference K/Na ratios, such as 1:1, 1:2, 2:1, and 3:1? If not, can the authors comment on the role of the relative contents of K and Na?
2. Some more details regarding the DFT calculations may be needed. For example, for typical supercells, the numbers of the K, Na, Nb, and O atoms included in the supercell should be clearly indicated in the text (or the supplementary materials). The size of the supercell should be also explicitly indicated. Also, Line 431 indicates a 1x8x7 mesh but Line 433 states a 1x1x9 cell – if the mesh was used for the supercell, the k-points set up may be problematic. Please clarify.
3. Liu et al [Journal of the American Ceramic Society, 97, 4019-4023 (2014)] reported their first-principles studies of K_{1-x}Na_xNbO₃ solid solutions, regarding the structures and the enhanced piezoelectric response. This work should be cited.
4. Line 344: (K_xNa_{1-x})NbO_{3-z} should be (K_xNa_{1-x})_yNbO_{3-z}.

Response letter (No. NCOMMS-21-48697)

The changes made to the manuscript are highlighted in yellow, which are also included here, when possible, given underlined in red.

Referee #1:

General Comment: *M. Waqar et al. demonstrate a small-signal piezoelectric coefficient of ~1900 pm/V at 1 kHz, and a large-signal electromechanical strain of 5.6 % in non-stoichiometric potassium sodium niobate epitaxial thin films with planar faults. Some of the authors have introduced a similar enhancement of the piezoelectric properties already in a 2020 science (DOI: 10.1126/science.abb3209) and nature communications (<https://doi.org/10.1038/s41467-021-231>) publications for NaNbO₃. In the current manuscript the authors focus on explaining the mechanism, utilizing (K,Na)NbO₃ thin films as an example.*

The authors find that the density of planar faults can be controlled by tuning the (K+Ka)/Nb ratio. With increasing planar fault concentration, an increase of the small- and large-signal piezoelectric response is reported. The enhancement of the piezoelectric behavior is prominent at low frequencies and is highly non-linear with the application of the electric field. The piezoelectric and strain performance is considerably enhanced in comparison to conventional thin films made from KNN or PZT. The authors characterize their films using a combination of advanced transmission electron microscopy, macroscopic spectroscopic analysis (impedance measurements and XPS) and DFT calculations. In summary, the authors explain the high electromechanical strain by self-trapped electrons and highly distorted oxygen octahedra in the vicinity of the self-assembled planar faults.

Innovation in strategies to enhance the piezoelectric performance beyond conventional PPT and MPB approaches are important for future thin film and bulk piezoelectric materials in terms of sustainability. In this frame the current manuscript is interesting and timely. However, the mechanism of the piezoelectric enhancement, the role of intrinsic piezoelectric properties and domain walls is unclear. To make the manuscript understandable and explain the mechanism to a broader audience this needs to be clarified, the core-message of the mechanism needs to be worked out more clearly and the impact of the work for the broader community needs to be explained. In the current form, I cannot recommend the manuscript for publication. I particularly suggest that the authors focus on the following points:

Response: We thank the referee for evaluating our work and for providing detailed and constructive feedback. Major revisions have been made to the manuscript in reference to the comments of the referee. We have categorized parts of manuscripts under different headings and rearranged the content (e.g., spectroscopic analysis is moved before the polarization analysis) to provide a smooth flow and clear understanding for the readers. We believe that the revisions made to the manuscript deal well with the concerns raised by the referee, and hope that the current version bears the required quality for publication in Nature Communications.

Comment 1: *The mechanism is unclear. According to my understanding the huge strain somehow originates mainly from the pillars formed by the planar faults. Can the authors please be crystal clear: What is the interplay between local elastic strains, polaronic charges, oxygen octahedra, polarization orientation of the surrounding domains and the applied electric field?*

Response: We appreciate referee's comments regarding the mechanism of piezoelectric enhancement. The correlation among local strain, oxygen octahedral distortion, localized charge and their interplay with polarization orientation has been summarized in the revised manuscript.

Pg. 9, Ln 208:

~~This indicates that the oxygen rumpling caused by the OODs at the PFs and the resulting formation of in-plane dipoles strongly affect the polarization states in the adjacent region. The PFs also cause a local in-plane strain (Fig. 2f, Fig. S8) which, in combination with the OODs, results in a symmetry-lowering distortion in the vicinity as suggested by local symmetry analysis^{20,21} (Fig. S7 and S8).~~ **To further visualize the effect of PFs on the domain structure and local symmetry in PF-KNN film, we performed large-area polarization mapping (See Supplementary Section IV). Based on the results, the local strain and built-in electric field present at the dense population of PFs appear to hinder the formation of long-range ferroelectric ordering resulting in nano-regions of complex polarization states enclosed within the PFs. Furthermore, the local chemical and structural heterogeneity at the PFs result in low symmetry phases (See Supplementary Section IV and Methods) in the vicinity which provide a continuous polarization rotation path from the in-plane direction near the PFs to the out-of-plane direction away from the PFs (Fig. S14 and Fig. S15). These observations show that the interplay between OODs and the localized charge at the PFs result in the formation of in-plane dipoles which strongly affect the polarization states in the adjacent region.**

Furthermore, the piezoelectric enhancement mechanism has also been summarized in the revised manuscript as follows:

Pg 11, Ln 238:

“Mechanisms for Giant Electromechanical Strain.

Based on the results, the structural and polarization inhomogeneity caused by the PFs in the PF-KNN thin film can be schematically illustrated, as shown in Fig. 3a. The tail-to-tail polarization at the PFs suggests that the local charge redistribution play a major role in the observed polarization orientations in the vicinity of the PFs. Fig. 3b and 3c depict the structural distortions adjacent to the PFs showing large variations in Nb-O bond lengths along with the in-plane dipole, as calculated by DFT. Large off-centering of Nb ions relative to the oxygen octahedra can be observed immediately adjacent to the PF due to the reduced ionic charge of Nb²⁺. This corresponds to a dramatically enhanced intrinsic electromechanical response including remarkably large lattice polarization adjacent to the PF interface based on the theoretical estimation (see Supplementary Section V). However, our analysis suggests that the intrinsic piezoelectric response cannot account for the observed giant electric-field-induced-strain without the involvement of extraordinarily large electrostrictive effects or other extrinsic factors. ~~This corresponds to an enhanced polarization. However, our analysis also suggests the involvement of other extrinsic factors in the observed large electromechanical response (see Supplementary Section V).~~ Polarization rotation between different crystallographic directions and lattice-shearing-induced phase transformations can result in a large electric-field-induced strain in the vicinity of PFs^{20,26,27}. However, under an applied electric field, the reorientation of the in-plane component of polarization in the vicinity of PFs further involves a competition between the applied out-of-plane electric field and the internal in-plane dipole caused by the localized charge and the OODs. A polarization reorientation as shown in Fig. 3d, hence, requires the destabilization of electrical and elastic boundary conditions at the PFs. ~~We propose that this can be facilitated and mediated by the migration of polaronic charge resulting in the~~

out-of-plane distortion of oxygen octahedra and the ionic displacements due to the variation of the Nb valence at the PF interface. When the field is removed, the original structural configuration reinstates the competing Coulombic repulsion and electron-lattice interactions at the PF interface. In this phenomenological model the polarization-rotation-induced large ionic displacements at the PFs mediated by polaronic charge migration causes the huge reversible electric field induced strain in the vicinity of PFs (Fig. 3e). Based on the experimental evidence provided in the next section, we argue that this polarization reorientation is facilitated and mediated by the electric-field-induced migration of localized polaronic charge along the PFs. The corresponding charge redistribution leads to large ionic displacements in association with the variation of the Nb valence and out-of-plane distortion of oxygen octahedra resulting in a large electric-field-induced strain in the vicinity of PFs. When the field is removed, the original structural configuration reinstates the competing Coulombic repulsion and electron-lattice interactions at the PF interface. Based on this model, we conclude that the large electromechanical response primarily originates from the vicinity of the PFs, and hence, a larger electric-field-induced strain is expected to result from the thin film with higher density of PFs. This is in accordance with the experimental results shown in Fig. 1a”

Comment 2: In the summary (page 15, line 334) the authors explain, that “planar faults provide an intermediate pathway for polarization switching”. No experimental evidence / discussion in the manuscript is available to support this statement. Experimental evidence should be provided, or the statement should be removed.

Response: The statement has been changed to the following:

Pg 18, Ln 394

~~The voltage, temperature, and frequency dependent electric field induced strain behaviors suggest that the ionic displacements and polaron migration near the planar faults provide an intermediate pathway for polarization switching and resulting large electric field induced strain.~~ “The voltage, temperature, and frequency-dependent electric-field-induced strain behaviors suggest that the polaronic charge migration along the planar faults facilitates the large ionic displacements associated with the polarization reorientation under an applied electric field. The resulting out-of-plane shearing and elongation of unit cell in the vicinity of planar faults facilitated by the structural heterogeneity results in the large observed electric-field-induced strain as supported by the theoretical calculations.”

Comment 3: Does the piezoelectric response change as a function of time? Many PPT and MPB compositions have the advantage of a low fatigue. How are the novel materials the authors are suggesting performing in comparison?

Response: We thank the referee for the insightful comments. The focus of the current work is to characterize the defect structure of the film in relation to the giant electromechanical response and to identify the structural features and mechanisms responsible for the observed behavior on the atomic scale. A more detailed study of other performance indicators from the point of view of practical applications of the films such as aging, cycling stability, fatigue resistance, etc, is underway through a new research project that we have secured recently.

Comment 4: What is the domain structure of the thin film and how do the interfaces formed by the planar faults impact the domain structure? Can experimental evidence (i.e. PFM) be provided? The authors discuss that tail-to-tail polarization configurations form, but they do not mention anything regarding domain morphology or the density of domain walls in the vicinity of the planar faults.

Response: We thank the referee for raising this good question. Accordingly, we have added further discussion on the domain morphology under the heading “Polarization Mapping and Domain Structure.” as also given in the response of Comment 1. To visualize the domain structure of the film, we performed polarization mapping on a large area in a scanning transmission electron microscopy study which is widely used as a reliable method to observe the domain structure. [*Nature Communications*, 2013, 4(1), 1-9., *Nature Communications*, 2020 11(1), 1-9]. The result is shown here and is added to the supplementary information as Fig. S15.

As shown in Fig. S15, long-range ferroelectric order is absent in the film and nano-regions of complex polarization states are observed enclosed within the PFs. Compared to conventional micro- or nano-domains observed in ferroelectrics with MPB, the polarization orientations found in PF-KNN film are complex in nature, showing continuous rotation from the in-plane direction near the PFs to the out-of-plane direction away from the PFs. Compared to the MPB ferroelectrics, while the composition and microscopic structural mechanism are different, the complexity of domain structure obtained here realizes continuously rotating polarization states, generally resembling other ferroelectric systems with competing elastic and electrostatic gradient energies. [*Nature*, 2019, 568(7752), 368-372., *Science*, 2019, 366(6464), 475-479, *Nature communications*, 2021, 12(1), 1-8.]

Fig. S15 Local polarization analysis and domain structure of PF-KNN thin film. In-plane polarization vector $\delta_{Nb,xy}$ map overlaid on inverted ABF image of thin-film plan-view where white dotted lines show the planar faults. Scale bar equals 2 nm. b. and c. represent the distribution of polarization magnitude and angle, respectively.

PFM results from PF-KNN disclose no detectable domain patterns as shown in Fig. R1, which supports the absence of long-range ferroelectric order in the films. Due to the limited resolution of the PFM technique compared to the atomic-scale features observed in the sample, STEM results provide clearer and more reliable information regarding the domain structure of the film as can be observed from the results presented in Fig. S14 and Fig. S15, in the revised version of the manuscript and SI.

Fig. R1 a) Vertical and b) lateral piezoforce microscopy (PFM) images from PF-KNN film.

Comment 5: *How do domain walls contribute to the piezoelectric response? In MPB compositions contributions from domain walls are dominating. Since the authors discuss their work in this context, I suggest that extrinsic / intrinsic contributions to the piezoelectric performance are discussed, to make it easier for the broader community to understand the mechanism the authors are suggesting.*

Response: As detailed in the response to the Comment 4, the domain structure of the PF-KNN film is different from conventional ferroelectrics with MPB. In order to bring further clarity to this issue, we have provided more details regarding the polarization orientations and symmetry of PF-KNN films in Supplementary Section IV and Fig. S14, apart from Fig. S15:

“The polarization vector mapping from a comparatively larger area is shown in Fig. S14. The polarization vectors represented by the shades of red show polarization with large in-plane component and the ones represented by the shades of blue represent polarization with a small in-plane component. It is evident from the STEM result shown in Fig. S14 that matrix perovskite KNN region (away from the PFs) shows an out-of-plane polarization and a small in-plane component of polarization vector which conforms with a tetragonal symmetry as is also supported by the XRD results in Fig. S4. However, the lattice in the vicinity of PFs has a larger in-plane component with a local symmetry distortion leading to lower symmetry rhombohedral and orthorhombic phases with possible existence of monoclinic bridging phases (Methods). Further analysis of polarization magnitude and angle given in Fig. S15 show continuous rotation between different polarizations.”

Fig. S14 In-plane polarization vector $\delta_{Nb,xy}$ map overlaid on inverted ABF image of thin-film plan-view where white dotted lines show the planar faults. Scale bar equals 1 nm. A tetragonal phase is observed in the matrix perovskite KNN phase whereas low symmetry rhombohedral and orthorhombic phases are observed in the vicinity of the PFs with possible presence of bridging monoclinic phases to facilitate polarization rotation between the PFs and the matrix.

The observed low symmetry phases merely act as a bridge for continuous polarization rotation between the planar faults and the matrix. No distinct phase boundaries or domain walls are observed. Indeed tail-to-tail polarization configuration observed at the planar faults in fact resembles charged domain walls. However, these planar faults can be considered as a special type of domain walls, where 180° domains are stabilized by locally accumulated strain and charge at the structural disorder i.e., planar faults which cannot be mobilized under electric field. Accordingly, the conventional mechanisms of domain switching and domain wall motion may or may not be applicable in this particular case. This is also evident from the unconventional voltage and frequency-dependent electromechanical response provided in Fig. 1d and Fig. 1e. The charge-migration-mediated switching mechanism at the planar faults complies well with the observed electromechanical behavior.

Comment 6: *The nonlinearity in the piezoelectric response can also originate from ferroelectric domain wall contributions (i.e. work by D. Hall: <https://link.springer.com/article/10.1023/A:1017959111402>). The authors utilize the nonlinearity of the piezoelectric properties to corroborate the polaron migration model. Can the nonlinear behavior also be explained by domain wall contributions? In general a more extended discussion on the contributions of domain walls to the piezoelectric performance is required in the manuscript.*

Response: We thank the reviewer for the useful comment. We have added the relevant discussion in the manuscript

Pg 15, Ln 335:

“In the literature, frequency-dependent piezoelectric non-linearities are attributed to different underlying mechanisms. For example, the pinning of non-180° domain walls by randomly distributed defects results in a logarithmic increase in the piezoelectric response with the decreasing frequency²⁸. In ferroelectric heterostructures or composites, the Maxwell-Wagner relaxation might be observed due to the differences in conductivities of the individual components^{29,30}. In some cases, piezoelectric relaxation may also be observed at very low frequencies due to the irreversible movement of conductive domain walls under external electric field³¹. While the presence of such mechanisms cannot be ruled out completely in PF-KNN film, they cannot account for more than an order of magnitude increase in the $d_{33,f}^*$ at low frequencies observed in our case. A low-frequency relaxation of electrostrictive response of the same order has been observed in oxygen-deficient oxides^{32,33}. However, we did not observe the presence of such large amounts of oxygen vacancies in our sample (see Supplementary Section VI). The frequency dependence of $d_{33,f}^*$ can, in fact, be well fitted with the non-ideal Debye equation suggesting that the piezoelectric relaxation should be associated with the reorientation of charged defects (polarons) under the alternating field^{34,35} (Fig. S17). Since the relaxation frequency of small polarons is in the range of 10^0 - 10^3 Hz¹⁴, the fitted value of piezoelectric relaxation frequency (~160 Hz) is in the expected range.”

***Comment 7:** The authors have published and discussed on the effect of planar faults in NaNbO₃. In the current manuscript they are using (K,Na)NbO₃. Can the authors explain why the effect is enhanced in (K,Na)NbO₃ in comparison to NaNbO₃? The authors state that “the higher electromechanical strain can be attributed to the introduction of K..” (line 115, page 5), the mechanism, however, is not explained.*

Response: We have detailed the mechanism in Supplementary Section 1:

“Secondly, the density of PFs in Fig. 1a (0.13 nm^{-1}) is close to that reported earlier in NPR-NNO, however, it shows a higher maximum $d_{33,f}^*$ of 1347 pm V^{-1} (at 90 kV cm^{-1}) at 1 kHz compared to that of 1098 pm V^{-1} (at 125 kV cm^{-1}). The former also shows an electric-field-induced strain of ~1.2 % at 90 kV cm^{-1} compared to ~0.5 % achieved in the latter at the same electric field. Hence, the chemical composition and the induced PFs play a significant role in the observed electromechanical response. In the literature, it is established that (K,Na)NbO₃ solid solutions are ferroelectric in nature and show superior piezoelectric properties compared to either of the parent components i.e., NaNbO₃ or KNbO₃.¹ Bulk NaNbO₃ has an antiferroelectric phase, without piezoelectric property. Due to its bigger size, the substitution of Na by K atoms in the niobate system disfavors antiferroelectric NbO₆ tilts and rotations while stabilizing the ferroelectric phase^{2,3}. Moreover, the geometrical frustration driven by Na/K size mismatch cause local fluctuations in the Nb-O bond lengths and tilt patterns which result in the softening of lattice and enhanced polarization rotation under external electric field^{4,5}. This largely improves the ferroelectric and piezoelectric response⁶, and could explain the higher electromechanical response in PF-KNN film compared to NPR-NNO film, even in the case with similar defect density.”

***Comment 8:** Why do the planar faults self-assemble into a nanopillar*

Response: Fig R2a given below shows a plan-view HAADF-STEM image of PF-KNN thin film showing an isolated pillar formed by planar faults. It is interesting to see that the planar faults step into different lattice planes (highlighted by an oval) or change the direction of propagation (highlighted by a triangle) after a critical length i.e., every 2- or 3-unit cells. We believe this critical length is related to the elastic strain associated with the planar fault.

According to our experimental results and DFT calculations, the interplanar distance between two adjacent NbO planes at the planar faults interface (2.59 \AA) is larger than half of the unit cell (2.01 \AA) due to charge redistribution and modified Coulomb interactions (Fig. S6). This is also shown in Fig. R2b where ϕ shows the difference between the interplanar distance of NbO planes at the faults and half the unit cell. This increased interplanar distance results in large elastic strain around the faults as shown in Fig. 2e. The elastic strain destabilizes the planar fault after a critical length. In addition to this, based on Wang et. al's work (*Proceedings of the National Academy of Sciences*, (2018)115(38), 9485-9490), planar defects which are close to each other tend to communicate and merge in order to minimize their surface energies. Based on these analyses, we believe that the observed morphology of planar faults is the result of the minimization of both the elastic strain as well as the surface energy of planar faults. Growth mechanism and precise morphological control are spared for future work.

Fig. R2 a) Plan-view HAADF image of PF-KNN thin film showing a pillar formation b) Schematic showing ϕ which is the difference between interplanar distance of NbO planes at the faults and half the unit cell.

Comment 9: *The localized charges seem to play a prominent role in the mechanism for the piezoelectric enhancement. Can the authors provide experimental proof for the existence of the localized charges? Techniques such as conductive Atomic Force Microscopy should be used to proof the existence of such charge carriers.*

Response: In our work, we have provided experimental proof of the localized charges at the planar faults using electron energy loss spectroscopy and impedance spectroscopy analysis. Furthermore, our experimental results are well supported by the first-principles calculations. Since the planar faults are densely populated, the distance between two planar faults (1 nm- 5 nm) is smaller than the resolution of currently available AFM techniques.

Comment 10: *Temperature-dependent displacement (Figure 4b) and temperature-dependent conductivity (Figure 4c) on PF-KNN thin films were used to support the polaron model suggested by the authors. Were similar experiments carried out on the stoichiometric KNN thin film (free of planar faults). Which values for the activation barrier were found for stoichiometric KNN?*

Response: We thank the referee for the valuable comment. The relevant experimental results and discussion have been added to the revised manuscript and SI:

Pg 8, Ln 178:

“Furthermore, the AC impedance of PF-KNN film shows a low frequency (~100 Hz) relaxation with an activation energy, E_a , of 0.33 eV indicating the existence of electrons as localized charges possibly in the polaronic form and not as itinerant charge carriers. This is also reflected in the AC conductivity analysis where the E_a of conduction in PF-KNN film is around 0.41 eV in comparison to the 0.02 eV obtained for stoichiometric KNN film (Fig. S11), which also confirms the presence of localized charged carriers in PF-KNN (See Supplementary Section III for detailed discussion).”

Fig. S11 AC conductivity analysis of KNN and PF-KNN films. a, Variation of AC conductivity with the frequency measured at different temperatures for KNN film. Symbols represent raw data points and solid lines show fitting by Jonscher’s power law. b, Calculation of activation energy from dc conductivity σ_{dc} obtained from (a). c, Variation of AC conductivity with the frequency measured at different temperatures for PF-KNN film. Symbols represent raw data points and solid lines show fitting by Jonscher’s power law. d, Calculation of activation energy from dc conductivity σ_{dc} obtained from (c).

Pg 14, Ln 308:

“The high-temperature electromechanical behavior of PF-KNN film is in stark contrast with the stoichiometric KNN film which shows a decrease in the $d_{33,f}^*$ at high temperatures and the electric-field-induced strain does not follow the Arrhenius law (Fig. S16). This further underpins the presence of fundamentally different physical mechanism responsible for large electromechanical strain in PF-KNN film compared to that of conventional piezoelectric thin films.”

Fig. S16 High-temperature electromechanical characterization of KNN thin film a, Variation in $d_{33,f}^*$ (and strain in the inset) measured at room temperature and 125 °C. b) Variation in the film strain with temperature under different external electric fields presented in Arrhenius formalism. All the piezoelectric measurements were done at 1 kHz.

Comment 11: Figure S10a: Was a similar experiment carrier out for stoichiometric KNN. Can a peak be observed? Relaxation in the frequency range of 10^0 - 10^3 Hz might have many reasons, including i.e. interfaces: <https://doi.org/10.1063/1.2757098>. Can the authors rule out such contributions, i.e. by performing measurements with different electrode materials or by comparing the results of KNN and PF-KNN?

Response: We thank the referee for the valuable comment. We have compared the results from KNN and PF-KNN as shown in Fig. S10, also presented below. Stoichiometric KNN film did not show relaxation in the frequency range of 10^2 - 10^3 Hz. Furthermore, the temperature had almost no effect on the impedance behavior of KNN film. This is completely different from the behavior of PF-KNN film. Based on the results, one can rule out contributions from the interface relaxation.

Fig. S10 AC impedance spectroscopy of KNN and PF-KNN film. a, Variation of the normalized imaginary part of the complex impedance (Z'') with the frequency measured for KNN film at the different temperatures. The inset shows a magnified view. b, Variation of the normalized Z'' with the frequency measured at the different temperatures for PF-KNN film. f_r represents the relaxation frequency of the charged defects. The inset shows the activation energy E_a calculated from impedance relaxation peaks measured at different temperatures. Relaxation time τ is calculated by using the formula $\tau = (2\pi f_r)^{-1}$.

Comment 12: Which role does the electric field distribution in the material play? I would expect that due to the presence of charge carriers at the planar faults the electric field is locally decreased (since the conductivity is locally enhanced), which would result in a decreased local piezoelectric response at the planar faults. Can the authors discuss the role of an inhomogeneously distributed electric field? I suggest that the authors discuss the work by Rojac et al. (i.e. DOI: 10.1002/adfm.201402963) and Geneko et al. (<https://link.aps.org/doi/10.1103/PhysRevB.97.144101>) in this context.

Response: We thank the referee for the valuable comment. The effects of inhomogeneous electric field due to local conductivity difference are usually characterized and discussed in terms of the Maxwell-Wagner effect (M-W). For example, in the work of Rojac et al. [*Advanced functional materials*, 2018, 25(14), 2099-2108], around two times enhancement is observed in the piezoelectric constant of BiFeO₃ at low frequencies due to a non-linear M-W effect. In our work, we observe ~50 times increase in $d_{33,f}^*$ at low frequencies which we have attributed to the relaxation of charge migration along the planar faults. Although the presence of the M-W effect in our samples cannot be ruled out completely, it might be overshadowed by the electrochemical relaxation at the planar faults. In the absence of an external electric field, the charge is localized at the planar faults which may increase the conductivity locally, however, under the external electric field, the redistribution of charge may locally facilitate polarization re-orientation in the vicinity of the planar faults. The electric field and polarization under external stimulus have been more thoroughly discussed in our previous work [*Nature Communications*, (2021) 12(1), 1-8].

As suggested by the referee, we have added relevant discussion in the revised manuscript:

Pg 15, Ln 334:

“In the literature, frequency-dependent piezoelectric non-linearities are attributed to different underlying mechanisms. For example, the pinning of non-180° domain walls by randomly distributed defects results in a logarithmic increase in the piezoelectric response with the decreasing frequency²⁹. In ferroelectric heterostructures or composites, the Maxwell-Wagner relaxation might be observed due to difference in the conductivities of the individual components^{30,31}. In some cases, piezoelectric relaxation may also be observed at very low frequencies due to the irreversible movement of conductive domain walls under external electric field³². While the presence of such mechanisms cannot be ruled out completely in PF-KNN film, they cannot account for more than an order of magnitude increase in the $d_{33,f}^*$ at low frequencies observed in our case.”

Comment 13: The authors find a tetragonal structure of the thin film (line 146, page 7). According to literature, pure KNN should be orthorhombic. Why is the film of the authors tetragonal?

Response: The film shows a tetragonal (P4mm) symmetry with the in-plane lattice parameters of $a = b = 3.926 \text{ \AA}$ and an out-of-plane lattice parameter $c = 4.0715 \text{ \AA}$. This is because of the epitaxial constraint imposed by the SrTiO₃ substrate ($a = b = c = 3.905$) on the film and is consistent with the previous reports (Jpn. J. Appl. Phys., Vol. 43, No. 9B (2004)).

Comment 14: The authors state that the symmetry lowering in the vicinity of the planar faults is monoclinic like (line 247, page 12). Based on which experiment do they conclude this?

Response: We thank the referee for pointing this out. The detailed symmetry analysis from STEM results has already been elaborated in the responses to Comment 4 and Comment 5. The results show a lowering of local symmetry in the vicinity of planar faults with continuous

polarization rotation, however, the confirmation of monoclinic phases is challenging from STEM analysis. Based on the theoretical calculations added to the revised manuscript (please see the response to Comment 2 of the Reviewer 3), the monoclinic Pm phase is locally obtained at the planar faults. The average structure experimentally obtained by the global HR-XRD is, however, tetragonal P4mm phase.

Comment 15: Figure 4b: Why is the temperature-dependence of the height change used? The unit “ln(nm)” is not straightforward at all. Wouldn't it make more sense to plot the strain (unitless) instead?

Response: We thank the referee for the suggestion We have revised Figure 4b with ln(strain) as the y-axis.

Fig. 4 Electrical characterization of defects in PF-KNN thin film a) Variation in $d_{33,f}^*$ (and strain in the inset) measured at room temperature and 125°C. b) Variation in the film strain with temperature under different external electric fields presented in Arrhenius formalism. c) Variation in the dc conductivity of PF-KNN film with the temperature under different external electric fields presented as Arrhenius plot. All the piezoelectric measurements were done at 1 kHz.

Comment 16: Figure 4c: The authors plot the DC-conductivity as a function of temperature? Where is this value coming from? I do not find frequency-dependent conductivity measurements, which allow to identify the DC conductivity in the supplementary (compare literature by Lunkenheimer et al. 10.1140/epjst/e2010-01212-5).

Response: The values of DC conductivity were obtained directly by using the source meter as stated in the Methods section. Based on the referee's suggestion we also derived DC conductivity from frequency-dependent conductivity using the Jonscher's power law and the results are shown in Fig. S11 and also provided in the response of Comment 10. Further details are also provided in Supplementary Section III:

“The dc conductivity (σ_{dc}) in Fig. S11 was calculated by fitting the σ_{ac} using the Jonscher's power law¹⁶:

$$\sigma_{ac}(\omega) = \sigma_{dc} + A\omega^n \quad \text{Eq. S2.}$$

where ω is the angular frequency and A, n are material dependent constants. This power law is termed as the Universal Dielectric Response and can be used to model the charge transport of localized charge carriers including hopping mechanism. The low frequency AC conductivity response of KNN film in the measured temperature range can only be loosely fitted using power law (Fig. S11a). However, in the contrary, the response of PF-KNN can be well fitted by power

law (Fig. S11c) indicating hopping dominated conduction mechanism which is characteristic of localized charge carriers.”

Comment 17: *The authors conclude that their mechanism “can be potentially applied to other materials” (page 16, line 341). Can the authors provide examples going beyond thin films?*

Response: This phenomenon can be further extended to polycrystalline thin films, and other nonstoichiometric compositions with ions of variable valence state. As shown in our group’s work, the structure and mechanisms have been observed in both NNO and KNN compositions. Self-assembled nanopillar-like defects can also grow along the oriented grains in polycrystalline films, leading to large piezoelectric response without the need for complex doping. It can be further extended to textured ceramics where such defects can be introduced using templated grain growth methods with nonstoichiometric compositions.

Referee # 2

General Comment: *The authors reported a record-high piezoelectric coefficient of about 1900 picometer per volt at 1 kHz in non-stoichiometric potassium sodium niobate epitaxial thin films ($K_x Na_{1-x} NbO_{3-z}$) with a high density of self-assembled planar faults. They also used first-principles modelling to elucidate the origin of this giant piezoelectric response in terms of atomic structure. The results are very impressive and deserve publication in some form. However, I have some comments for the authors to address:*

Response: We thank the referee for appreciating our work and for providing the valuable suggestions.

Comment 1: *Fig. S5 shows the optimized crystal structure of $(K_{0.3}Na_{0.7})NbO_3$ with one NaO/KO layer at the interface removed and then the two terminations are shifted by one-half of the supercell to have a "plane fault" (i.e. edge-sharing oxygen octahedra). Compared to the ideal stacking (panel a), the distorted structure (panel b) has lower energy. Fig. S6 shows that the Nb-O bond length at the interface increases from 2 Å to 2.6 Å. Presumably,*

Fig. S5 and S6 are the same DFT results. However, if one examines the two structures carefully, one finds that in Fig. S5 panel b, the NbO₆ octahedra at the interface have rotations and tilts, while in Fig. S6 panel b, the NbO₆ octahedra do NOT have any rotations and tilts and only Nb atoms are shifted along (001) direction, leading to polarization. Could the authors comment on why the optimized crystal structures in Fig. S5 and Fig. S6 are different?

Response: The optimized structures in Fig. S5b and Fig S6b are the same, however, Fig. S5b shows the structure along the [100] direction, whereas Fig S6b shows the structure along the [001] direction. A 3D view of this structure is provided in Fig. S5d, according to which there are no visible rotations of oxygen octahedra. To further make this point clear, we have provided the ball-and-stick models of oxygen octahedra viewed from different directions in Fig R3. The structure shown in Fig. R3a is along [100] direction and the one in Fig. R3b is a bit rotated to reveal the overlapped oxygen atoms. Fig. R3c shows the structure along the [001] direction. The shaded region highlights the planar faults. With the help of the horizontal dashed lines as reference guides, it can be understood from Fig. R3a that there is a small tilting or bending of horizontal oxygen planes (001 planes) at the planar fault which decreases going farther away from the fault interface. However, from Fig. R3b, it is obvious that such bending is absent in the vertical oxygen planes (010 planes). Hence, the incomplete tilting of NbO₆ observed here is different from the antiferroelectric or non-polar tilting of oxygen octahedron. Furthermore,

no visible rotation of NbO₆ octahedra is observed in Fig. R3b which is in line with the structure showed in Fig. R3c. Large distortions of octahedra can surely be observed which are commensurate with the large polarization especially at the interface.

Fig. R3 Ball-and-stick model of oxygen octahedra in PF-KNN viewed from a) [100] direction, b) slight rotation away from [100] direction and c) [001] direction.

Comment 2: A related comment: Line 154-155, the authors wrote "The larger size of the K ions compared to Na constrains the O positions, disfavoring octahedral rotation and favoring polar distortions". However, from Fig. S5, it seems that there are NbO₆ oxygen octahedral rotations and tilts, in particular at the interface (interestingly, such oxygen octahedral rotations and tilts are absent in Fig. S6). Could the authors also comment on that?

Response: This has been discussed in the response to Comment 1.

Comment 3: In the simulation, the K atoms are highly ordered, which also form a plane, similar to "plane fault". But experimentally, K and Na atoms are randomly alloying. Does the result (Fig. S6) depend on how close the "K plane" is to the interface, whether the two "K planes" are symmetric with respect to the interface and whether the K atoms are ordered or randomly distributed?

Response: We thank the referee for the comment and question. We agree with the referee that a random disorder of K and Na is possible in the crystal lattice of the sample in real conditions. We did calculate structures with different positions of K-planes in the supercell such as given in Fig. R4 and the results reported in the work were selected from among them. Irrespective of the position of the K-layer, the results showed no substantial variations in the observed structure (compare Fig. R4a and Fig. R4b). Hence the basic structural model is mostly unaltered by the position of the K atoms.

Fig. R4 Relaxed structures for PF-KNN film calculated from the DFT with K-Plane at different positions relative to the PF.

Comment 4: *The DFT calculations focus on the crystal structure but do not show the electronic structure. Could the authors provide the detailed density of states of the $(K_{0.3}Na_{0.7})NbO_3$ thin films with "plane faults" as well as the one without "plane faults"?*

Response: We thank the referee for the useful comment. We have added the DOS information in the revised manuscript as follows:

Pg 22, Ln 510, Methods:

“The density of states (DOS) was calculated for both KNN and PF-KNN films. In case of KNN, the structure was built from $NaNbO_3$ with a $5 \times 1 \times 1$ supercell by replacing one Na atom by a K atom.”

Fig. S9 Density of states (DOS) calculated for **a**, KNN and **c**, PF-KNN film. **b**, and **d**, show the enlarged view of (a) and (c) respectively, for K and Na. The Fermi level is located at $E=0$ as highlighted by dotted lines in the plots. In KNN, the valence band is dominated by O-states with contribution from Nb, whereas the K- and Na-states show insignificant contribution. The conduction band, however, is dominated by Nb-states with some contribution from O. Based on the DOS given in (a), KNN shows a semiconducting behaviour. With the addition of planar faults in KNN, the whole DOS is shifted to the lower energy compared to the pristine KNN structure as shown in (c). The Fermi level for PF-KNN is in the conduction band showing the n-type nature of the system due to the excess electrons at the PF. This further supports the lowering of ionic charge of Nb at PFs due to charge redistribution. Overall, PF-KNN shows metallic behaviour.

Comment 5: A related comment: Line 189-196, the authors wrote "confirming the lowering of Nb valence at the PF interface", "This variation is likely facilitated either by band bending or electron transfer from neighboring oxygen atoms", "the nominally empty 4d orbitals of Nb atoms become partially occupied at the PF interface", "these electrons exist as localized charges possibly in the polaronic form and not as itinerant charge carriers". Do the DFT calculations support the above claims, in particular from the electronic structure?

Response: The lowering of Nb valence is confirmed from the DFT calculations, as given in the response of Comment 6 (Fig. S9), and also based on Fig. S8. The localization of charge carriers is a conclusion based on the fact that the DOS calculations show that the PF-KNN sample is metallic, however, the PF-KNN sample has its conductivity comparable to that of the KNN sample at room temperature (Fig. S11) showing that the excess charge carriers are localized.

Comment 6: In the DFT calculations, presumably the $(K_{0.3}Na_{0.7})NbO_3$ thin films with "plane faults" are insulating. If that is the case, can the authors directly calculate d_{33} and compare

it to the experiment? In addition, bulk (K_{0.3}Na_{0.7})NbO₃ without "plane faults" is also insulating. Can the authors also calculate its d_{33} and compare it to the one with the "plane faults"?

Response: Unfortunately, we are not able to calculate d_{33} directly. This is because the electronic structure obtained in our DFT calculations is not insulating. This is seen in the DOS for planar fault KNN, as shown in Fig. S9. As discussed, carrier localization is needed to obtain an insulating state in accordance with the experiment. The metallic nature of the DFT result precludes the direct calculation of d_{33} . Therefore, we are not able to compare DFT results for the KNN with and without planar faults.

Referee # 3

General comment: *The authors report their finding of a high piezoelectric coefficient, approximately 1.9 nm/V, and a large strain of 5.6% induced by electric fields in non-stoichiometric potassium sodium niobate [(KxNa1-x)yNbO3-z] thin films. The authors further provide an interpretation, based on experimental measurements and first-principles calculations of atomic structures (with an emphasis of the role of planar faults), polarization, and electronic properties, about the large strain induced by electric fields. As piezoelectrics with large electromechanical response is of great interests in industry, this work have potentially engineering applications. The conclusions are supported by both the experimental data and DFT calculations. The manuscript is also well-written. Thus, I would recommend an acceptance provide that the authors can address the following points in a revised manuscript:*

Response: We are grateful to the referee for his/her valuable time and for the appreciation of our work.

Comment 1: *The PF-KNN films [(KxNa1-x)yNbO3-z] are the focus of this work. Three films, (K0.26Na0.74)0.52NbO3-z, (K0.24Na0.76)0.85NbO3-z, and (K0.26Na0.74)0.60NbO3-z were obtained for experimental analysis while DFT calculations focused on K0.3Na0.7NbO3. These films all have a similar relative ratio (approximately 1:3) of the K and Na contents. Did the work involve other films with difference K/Na ratios, such as 1:1, 1:2, 2:1, and 3:1? If not, can the authors comment on the role of the relative contents of K and Na?*

Response: We thank the referee for the valuable comment. The relative amount of Nb in the films has indeed been the topic of discussion in this work. We did discuss the role of K in the film as given in the response of Comment 7 of Reviewer 1. The role of K relative to Na is related to the relatively larger size of K. We expect the elastic strain to increase at the planar faults with the addition of K which might result in the modification of growth as well as the morphology of planar faults as discussed in the response of Comment 8 of Reviewer 1. Further experimental investigations for the relative amount of K and Na will be conducted in the future.

Comment 2: *Some more details regarding the DFT calculations may be needed. For example, for typical supercells, the numbers of the K, Na, Nb, and O atoms included in the supercell should be clearly indicated in the text (or the supplementary materials). The size of the supercell should be also explicitly indicated. Also, Line 431 indicates a 1x8x7 mesh but Line 433 states a 1x1x9 cell – if the mesh was used for the supercell, the k-points set up may be problematic. Please clarify.*

Response: We appreciate referee's valuable comment. In fact, we made an error in the statement of the supercell size, which is now corrected. Thank the referee for pointing out our mistake. We have rechecked all the calculations and added more details regarding the atomic composition and dimensions of the supercell as well as regarding the calculations in the revised manuscript:

Pg 22, Ln 499, Methods

~~We built a supercell of 1×1×9 and removed one NaO/KO layer at the interface then shifted one half of the supercell (1/2a,1/2b,0). For distorted planar faults, we shifted one half of the supercell the supercell (1/2a+α,1/2b+α,0).~~ **We built a supercell of 11×1×1 and removed two NaO/KO layers at the interface then shifted one half of the supercell (1/2a,1/2b,0) in accordance with the structure presented in Fig. 2e with the composition of 2 K, 7 Na, 11 Nb**

and 31 O atoms. For distorted planar faults we shifted one half of the supercell $(1/2a+\alpha, 1/2b+\alpha, 0)$. The new optimized structure at the planar fault has lattice constant of $a=40.68\text{\AA}$, $b=3.905\text{\AA}$ and $c=4.05\text{\AA}$ with monoclinic Pm symmetry.

***Comment 3:** Liu et al [Journal of the American Ceramic Society, 97, 4019-4023 (2014)] reported their first-principles studies of $K_{1-x}Na_xNbO_3$ solid solutions, regarding the structures and the enhanced piezoelectric response. This work should be cited.*

Response: We thank the referee for the useful suggestion. We have added the reference in the revised manuscript as Reference no. 28.

***Comment 4:** Line 344: $(K_xNa_{1-x})NbyO_{3-z}$ should be $(K_xNa_{1-x})_yNbO_{3-z}$.*

Response: We thank the referee for pointing out the mistake. We have revised it accordingly.

REVIEWER COMMENTS

Reviewer #1 (Remarks to the Author):

The authors have answered all of my questions. The mechanism, particularly the contributions of ferroelectric domain walls to the giant piezoelectric response is now more clear.

Regarding my comment 17: To make the mechanism and the manuscript accessible to a broader readership, I suggest that the authors mention processing approaches that can be utilized to introduce defects in the manuscript. Texturing might be an option, although the introduced defects might be of micrometer size and thus unsuitable for the suggested mechanism. Other recent approaches resulting in nanoscale or atomic size defects worth mentioning are dislocations (10.1126/science.abe3810) or precipitates (10.1002/adma.202102421).

Reviewer #2 (Remarks to the Author):

The authors addressed most of my comments, which I appreciate.

However, about comments #5 and #6 (charge localization), the current answers are not satisfactory. Their DFT calculations find a metallic state (in particular a peak emerges at the Fermi level), while experimentally the films exhibit insulating behavior. Then the authors argued that "carrier localization is needed to obtain an insulating state in accordance with the experiment.". This argument does not provide anything deep. It basically says that the DFT calculation fails to reproduce the experimentally observed insulating state. But why is "carrier localization", if it does exist, missing in the standard DFT calculations? If the charges are indeed in a polaronic form and are rendered immobile, then this in principle could be captured by DFT calculations, since it is due to structural distortions. Could it be due to correlation effects on Nd-d orbitals?

The authors did a detailed analysis on the structural properties. However, if the "carrier localization" does occur, it should also affect the crystal structure, no matter whether it is due to polaron or correlation effects.

I suggest that the authors perform some additional calculations (either using DFT+U or other methods) to reproduce an insulating state and then show whether the crystal structure at the planar faults may strongly depend on the insulating/metallic nature of the films.

Reviewer #3 (Remarks to the Author):

The authors have appropriately addressed all the points I raised in my previous review, and I would therefore recommend acceptance of the revised manuscript.

Response letter (No. NCOMMS-21-48697A)

The changes made to the manuscript are highlighted in **yellow**, which are also included here, when possible, given underlined in **red**.

Referee #1:

***General Comment:** The authors have answered all of my questions. The mechanism, particularly the contributions of ferroelectric domain walls to the giant piezoelectric response is now more clear. Regarding my comment 17: To make the mechanism and the manuscript accessible to a broader readership, I suggest that the authors mention processing approaches that can be utilized to introduce defects in the manuscript. Texturing might be an option, although the introduced defects might be of micrometer size and thus unsuitable for the suggested mechanism. Other recent approaches resulting in nanoscale or atomic size defects worth mentioning are dislocations (10.1126/science.abe3810) or precipitates (10.1002/adma.202102421).*

Response: We thank the referee for evaluating our revisions and for the useful suggestion. We have added the following statement to the revised manuscript along with the relevant references.

Pg 3 ln 58

In this regard, a large improvement in piezoelectric response has been observed in perovskite oxide ferroelectric crystals and ceramics by chemically and mechanically induced nanoscale defects such as vacancies, dislocations, and precipitates⁹⁻¹¹.

Referee # 2

General Comment: *The authors addressed most of my comments, which I appreciate. However, about comments #5 and #6 (charge localization), the current answers are not satisfactory. Their DFT calculations find a metallic state (in particular a peak emerges at the Fermi level), while experimentally the films exhibit insulating behavior. Then the authors argued that "carrier localization is needed to obtain an insulating state in accordance with the experiment.". This argument does not provide anything deep. It basically says that the DFT calculation fails to reproduce the experimentally observed insulating state. But why is "carrier localization", if it does exist, missing in the standard DFT calculations? If the charges are indeed in a polaronic form and are rendered immobile, then this in principle could be captured by DFT calculations, since it is due to structural distortions. Could it be due to correlation effects on Nd-d orbitals?*

The authors did a detailed analysis on the structural properties. However, if the "carrier localization" does occur, it should also affect the crystal structure, no matter whether it is due to polaron or correlation effects. I suggest that the authors perform some additional calculations (either using DFT+U or other methods) to reproduce an insulating state and then show whether the crystal structure at the planar faults may strongly depend on the insulating/metallic nature of the films.

Response: We thank the referee for the insightful discussion on the theoretical calculations. Our spectroscopic analyses presented in Fig. 2h and Fig. S8 show the existence of Nb *4d* electrons which are expected to contribute to the metallic conductivity at the planar faults. This is well supported by the DFT calculations. Consequently, the films show high conductivity and leakage at high electric fields as shown in Fig 4c. The charge localization is apparent at low electric fields where the films show a relatively lower, thermally activated conductivity (Fig. 4c) with low frequency relaxation (Fig. S10). We have proposed that this localization is related to the electron-lattice interactions due to the large structural distortions at the planar faults which modify the crystal structure in the vicinity. We agree with the reviewer that the electron correlation effects, in principle, could also localize the excess electrons. However, here we are dealing with an early *4d* transition element Nb for which the correlation effects may be smaller compared to that of an element in a *3d* system. In order to test this, we have done PBE+U calculations as suggested by the referee. Specifically, we did new GGA+U calculations with the Hubbard parameter set to a large value $U=0.5Ry$ (6.8 eV). The obtained results, also presented below, still show a metallic state, with the density of states similar to the calculation with $U=0$. This suggests that the carrier localization is unlikely to be caused by the Coulomb correlations further supporting our argument of carrier localization having a lattice character. Hence, our DFT calculations indeed support the experimental findings in this work. We also noted that Nb^{4+} oxides have a strong tendency to distort where these distortions are often complex with a tendency towards structural instabilities with materials having nominal Nb^{4+} valence and cannot be typically captured by simple charge disproportionation. An example is provided in one of our past works (Phys. Rev. Lett. 93, 216403, 2004), where the distortion of Nb in a simple ternary compound with a cubic structure required a large unit cell to describe. The structural distortions due to Nb^{5+}/Nb^{4+} valence variations may enhance the tendency towards charge localization due to lattice effects. Unfortunately, the theoretical illustration to describe polaron formation goes well beyond our computational capabilities, because of the prohibitive size of the supercell model that would be required. To present more meaningful discussion regarding the theoretical aspects of charge localization, we have added the following to the manuscript:

Pg 9 ln 190

In ferroelectric materials, such localization is caused by the self-trapping of electrons in potential wells created by local elastic distortions²³. Hence, in our case, the electronic charge is localized around the PFs due to the presence of an anisotropic elastic strain field caused by the OODs^{24,25}. Our Generalized Gradient Approximations for Coulomb interaction potential (GGA+U) show that the electron correlation has no significant effects on the electronic properties suggesting the electron localization to have purely lattice character (see Methods, Fig. S9). Accordingly, we conclude that the delicate balance between the elastic distortions and localized charge at the PFs govern the observed in-plane OODs in the PF-KNN film.

Pg 21 Ln 496

“The DOS show metallic behaviour at the planar interface which explains the large conductivity and leakage at high electric fields. However, the carrier localization at the planar faults due to lattice interactions presents a barrier to the conductivity at low electric fields having an activation energy of ~0.4 eV. To eliminate the possibility electron correlation effect at the planar fault, we conducted Perdew-Burke-Ernzerhof (PBE+U) calculations. Specifically, we performed GGA+U calculations with the Hubbard parameter set to a large value $U=0.5Ry$ (6.8 eV) as shown in Fig. S9. The obtained results still show a metallic state, with the density of states similar to the calculation with $U=0$. This suggests that the carrier localization is unlikely to be caused by the Coulomb correlations further supporting our argument of carrier localization having a lattice character.”

Fig. S9 Density of states (DOS) calculated for **a**, KNN and **c**, PF-KNN film. **b**, and **d**, show the enlarged view of (a) and (c) respectively for K and Na. The Fermi level is located at $E=0$ as highlighted by dotted lines in the plots. In KNN, the valence band is dominated by O-states with contributions from Nb, whereas the K- and Na-states show insignificant contributions. The conduction band, however, is dominated by Nb-states with some contribution from O. Based on the DOS given in (a), KNN shows a semiconducting behavior. With the addition of planar faults in KNN, the whole DOS is shifted to the lower energy compared to the pristine KNN structure as shown in (c). The Fermi level for PF-KNN is in the conduction band showing the n-type nature of the system due to the excess electrons at the PF. This further supports the lowering of the ionic charge of Nb at PFs due to charge redistribution. Overall, PF-KNN shows metallic behavior. e,f PBE-GGA+U calculation for PF-KNN structure with \$U=0.5\$ Ry (6.8 eV). The results show that the electron correlation has no significant effect on the electronic properties of PF-KNN.

Referee # 3

General comment: *The authors have appropriately addressed all the points I raised in my previous review, and I would therefore recommend acceptance of the revised manuscript.:*

Response: We thank the referee for evaluating our revisions and for the acceptance of the manuscript for publication.

REVIEWERS' COMMENTS

Reviewer #2 (Remarks to the Author):

The authors performed additional calculations. While they still fail to reproduce the insulating state, their new DFT+U calculations rule out the correlation effects as the source for charge localization, rendering the polaron mechanism more likely. The added text also makes this point more clear. With these revisions, I recommend the publication of this work in Nature Communications.